# Exact Verification of ReLU Neural Control Barrier Functions

**Hongchao Zhang**
Electrical & Systems Engineering
Washington University in St. Louis
St. Louis, MO 63130
hongchao@wustl.edu

**Junlin Wu**
Computer Science & Engineering
Washington University in St. Louis
St. Louis, MO 63130
junlin.wu@wustl.edu

**Yevgeniy Vorobeychik**
Computer Science & Engineering
Washington University in St. Louis
St. Louis, MO 63130
yvorobeychik@wustl.edu

**Andrew Clark**
Electrical & Systems Engineering
Washington University in St. Louis
St. Louis, MO 63130
andrewclark@wustl.edu

## Abstract

Control Barrier Functions (CBFs) are a popular approach for safe control of nonlinear systems. In CBF-based control, the desired safety properties of the system are mapped to nonnegativity of a CBF, and the control input is chosen to ensure that the CBF remains nonnegative for all time. Recently, machine learning methods that represent CBFs as neural networks (neural control barrier functions, or NCBFs) have shown great promise due to the universal representability of neural networks. However, verifying that a learned CBF guarantees safety remains a challenging research problem. This paper presents novel exact conditions and algorithms for verifying safety of feedforward NCBFs with ReLU activation functions. The key challenge in doing so is that, due to the piecewise linearity of the ReLU function, the NCBF will be nondifferentiable at certain points, thus invalidating traditional safety verification methods that assume a smooth barrier function. We resolve this issue by leveraging a generalization of Nagumo's theorem for proving invariance of sets with nonsmooth boundaries to derive necessary and sufficient conditions for safety. Based on this condition, we propose an algorithm for safety verification of NCBFs that first decomposes the NCBF into piecewise linear segments and then solves a nonlinear program to verify safety of each segment as well as the intersections of the linear segments. We mitigate the complexity by only considering the boundary of the safe region and by pruning the segments with Interval Bound Propagation (IBP) and linear relaxation. We evaluate our approach through numerical studies with comparison to state-of-the-art SMT-based methods. Our code is available at https://github.com/HongchaoZhang-HZ/exactverif-reluncbf-nips23.

## 1 Introduction

Safety is a critical property for autonomous systems, including unmanned ground, aerial, and space vehicles [1] and robotic manipulators [2]. The importance of safety has motivated extensive research into verification and synthesis of safe control strategies [3, 4, 5, 6, 7]. Control barrier function (CBF)-based algorithms [8] ensure safety by constructing a CBF that is nonnegative if the system is safe and synthesizing a control policy that ensures that the CBF is nonnegative for all time. Control barrier functions provide a high degree of flexibility, since any control policy that satisfies the control barrier function constraint is provably safe, and have been demonstrated in applications including

37th Conference on Neural Information Processing Systems (NeurIPS 2023).

robotic manipulation [9], vehicle cruise control [10] and space exploration [11]. Recently, CBFs that are parametrized by neural networks (*neural control barrier functions* or NCBFs) [12, 13, 14] have been proposed. This novel class of NCBFs is promising due to the universal representability of neural networks (enabling the encoding of complex safety constraints) and the efficiency of learning algorithms, with a comparison in Section A.15 exemplifying numerical difference. However, safety verification of NCBF-based control remains a challenging research problem.

In this paper, we consider the problem of verifying safety of NCBF-based control for nonlinear continuous-time systems. We focus on NCBFs represented by feedforward neural networks with ReLU activation due to their fast convergence in training shown in Section A.16 and widespread use in the safe control literature [12, 13]. The key challenge for this class of NCBFs is that most methodologies for safety verification are based on proving that the derivative of the barrier function is nonnegative at the boundary of the safe region, and hence the barrier function remains nonnegative for all time. Since the ReLU activation function is not continuously differentiable, this approach is inapplicable. We resolve this challenge and derive exact safety conditions for ReLU NCBFs by leveraging a generalization of Nagumo's theorem for proving invariance of sets with nonsmooth boundaries. Based on these conditions, we propose a new class of NCBF-based control policies that do not require the control policy and NCBF to be jointly trained, thus improving flexibility and tunability of the controller design. Our safety conditions also incorporate linear constraints on the control input, which may arise due to limits on actuation.

We propose an algorithm to verify that an NCBF satisfies our derived safety conditions, implying that any control policy that is constrained by the NCBF will be safe. Our approach first decomposes the NCBF into piecewise linear segments. In order to mitigate the complexity of this stage, we show that it suffices to consider linear segments at the boundary of the safe region, and further over-approximate the segments using Interval Bound Propagation and linear relaxation. After decomposing the NCBF, we verify safety by solving a set of nonlinear programs, which check the safety criteria on each linear segment as well as at intersections of the segments. We evaluate our approach through numerical studies, in which we verify NCBFs with our proposed approach and compare to state-of-the-art Satisfiability Modulo Theory (SMT) based methods.

**Related Work** Energy-based methods have been proposed to guarantee safety by ensuring that a particular energy function remains nonnegative. Barrier certificates for safe control were first proposed in [15]. More recently, CBFs have emerged as promising approaches to safe control, due to their compatibility with a wide variety of control laws [8, 16, 17, 5, 14, 12, 1]. Sum-of-squares (SOS) optimization and other techniques derived from algebraic geometry have been widely used for safety verification of polynomial barrier functions [15, 8, 18, 19, 17]. However, SOS-based approaches for polynomial CBFs cannot be applied directly to NCBF verification, since activation functions used in neural networks are not polynomial and may be non-differentiable.

Neural barrier certificate [20, 21, 22, 23] and NCBFs [12, 13, 14] have been proposed to describe complex safety constraints that cannot be encoded polynomials. Current work, including SMT-based methods [24, 21, 25] and mixed integer programs [26], verify safety by constructing a nominal control policy and proving that it satisfies the NCBF constraints. However, as we show in Section 5, the reliance on a particular control policy may lead to false negatives during safety verification. Another related body of work deals with the problem of verifying neural networks including SMT [27], output reachable set verification [28], polynomial approximations of the barrier function [29], verify input/output relationships [30, 31, 32] and ReLU neural networks focused verification [33, 34]. These methods, however, are not directly applicable to the problem of NCBF verification, which requires joint consideration of the neural network and the underlying nonlinear system dynamics. Methods based on input/output relationship can verify by encode dynamics and control policy with neural networks. However, with approximating error introduced, these methods are not directly applicable for exact verification. Piecewise linear approximations of ReLU neural networks have been used to develop tractable safety verification algorithms using linear [35] and SOS programming [36]. These approaches leads to sound and incomplete verification algorithms and have only be applied for discrete-time systems, whereas the present paper proposes exact verification algorithms for continuous-time systems.

**Organization** The remainder of the paper is organized as follows. Section 2 gives the system model and background on neural networks and notation. Section 3 presents the problem formulation and

exact conditions for safety. Section 4 presents our approach to verifying that an NCBF satisfies the safety conditions. Section 5 contains simulation results. Section 6 concludes the paper.

## 2  Model and Preliminaries

In this section, we first present the system model and definition of safety. We then give background and notations of feedforward neural networks.

### 2.1  System Model and Safety Definition

We consider a continuous-time nonlinear control-affine system with state $x(t) \in \mathcal{X} \subseteq \mathbb{R}^n$, control input $u(t) \in \mathcal{U} \subseteq \mathbb{R}^m$, and dynamics

$$\dot{x}(t) = f(x(t)) + g(x(t))u(t), \tag{1}$$

where $f : \mathbb{R}^n \to \mathbb{R}^n$ and $g : \mathbb{R}^n \to \mathbb{R}^{n \times m}$ are known functions. A control policy is a function $\mu : \mathbb{R}^n \to \mathcal{U}$ that maps a state $x$ to a control input $u$.

Safety of dynamical systems requires $x(t)$ to remain in a given region $\mathcal{C}$, which we denote as the safe region. We assume the safe region is given by $\mathcal{C} = \{x : h(x) \geq 0\} \subseteq \mathcal{X}$, for some function $h : \mathcal{X} \to \mathbb{R}$. Safety is related to the property of positive invariance, which we define as follows.

**Definition 1.** *A set $\mathcal{D} \subseteq \mathbb{R}^n$ is positive invariant under dynamics (1) and control policy $\mu$ if $x(0) \in \mathcal{D}$ and $u(t) = \mu(x(t)) \; \forall t \geq 0$ imply that $x(t) \in \mathcal{D}$ for all $t \geq 0$.*

We define a control policy $\mu$ to be *safe* if there is a set $\mathcal{D}$ such that (i) $\mathcal{D} \subseteq \mathcal{C}$ and (ii) $\mathcal{D}$ is positive invariant under dynamics (1) and control policy $\mu$.

One approach to designing safe control policies is to choose a function $b : \mathbb{R}^n \to \mathbb{R}$, denoted as a *Control Barrier Function (CBF)*, and let $\mathcal{D} = \{x : b(x) \geq 0\}$. In the case where $b$ is continuously differentiable, the following result can be used to guarantee positive invariance of $\mathcal{D}$.

**Theorem 1** ([8])**.** *Suppose that $b$ is a CBF, $b(x(0)) \geq 0$, and $u(t)$ satisfies*

$$\frac{\partial b}{\partial x}(f(x) + g(x)u) \geq -\alpha(b(x)). \tag{2}$$

*for all $t$, where $\alpha : \mathbb{R} \to \mathbb{R}$ is a strictly increasing function with $\alpha(0) = 0$. Then the set $\mathcal{D} = \{x : b(x) \geq 0\}$ is positive invariant.*

Theorem 1 implies that, if $b$ is a CBF, then adding (2) as a constraint on the control at each time step suffices to guarantee safety.

Recently, Neural Control Barrier Functions (NCBFs), in which the function $b$ is encoded by a neural network, have been proposed [12, 13, 14]. The advantage of NCBFs arises from the universal representability of neural networks, which allows them to realize a wide variety of safety constraints.

### 2.2  Neural Network Background and Notation

We give notations to describe a neural network (NN) with $L$ layers and $M_i$ neurons in the $i$-th layer. We let the input to the NN be denoted $x$, the output at the $j$-th neuron of the $i$-th layer be denoted $z_{ij}$, and the output of the network be denoted $y$. We let $\mathbf{z}_i$ denote the vector of neuron outputs at the $i$-th layer. The outputs are computed as

$$z_{ij} = \begin{cases} \sigma(W_{ij}^T x + r_{ij}), & i = 1 \\ \sigma(W_{ij}^T \mathbf{z}_{i-1} + r_{ij}), & i \in \{2, \ldots, L-1\} \end{cases}, \quad y = \Omega^T \mathbf{z}_L + \psi \tag{3}$$

where $\sigma : \mathbb{R} \to \mathbb{R}$ is the activation function. The input to the function $\sigma$ is the *pre-activation input* to the neuron, and is given by $W_{1j}^T x + r_{1j}$ for the $j$-th neuron at the first layer and $W_{ij}^T \mathbf{z}_{i-1} + r_{ij}$ for the $j$-th neuron at the $i$-th hidden layer. $W_{ij}$ has dimensionality $n \times 1$ for $i = 1$ and $M_{i-1} \times 1$ for $i > 1$. $W_i$ is an $n \times M_i$ matrix. In this paper, we assume that $\sigma$ is the ReLU function $ReLU(z) = \max\{0, z\}$. The output of the network is given by $y = \Omega^T \mathbf{z}_L + \psi$, where $\Omega \in \mathbb{R}^{M_L}$ and $\psi \in \mathbb{R}$. The $j$-th neuron at the $i$-th layer is *activated* by a particular input $x$ if its pre-activation input is nonnegative, *inactivated* if the pre-activation input is nonpositive, and *unstable* if the pre-activation input is zero.

A set of neurons $\mathbf{S} = (S_1, \ldots, S_L)$, with $S_i \subseteq \{1, \ldots, M_i\}$ denoting a subset of neurons at the $i$-th layer, is activated by $x$ if all of the neurons in $\mathbf{S}$ are activated by $x$ and all the neurons not in $\mathbf{S}$ are inactivated by $x$. A set of neurons $\mathbf{T} = (T_1, \ldots, T_L)$ with $T_i \subseteq \{1, \ldots, M_i\}$ is unstable by $x$ if all of the neurons in $\mathbf{T}$ are unstable by $x$.

For a given set $\mathbf{S}$, if $\mathbf{S}$ is activated by $x$, then the pre-activation input to each neuron and the overall output of the network are affine in $x$, with the affine mapping determined by $\mathbf{S}$ as follows. For the first layer, we define

$$\overline{W}_{1j}(\mathbf{S}) = \left\{ \begin{array}{ll} W_{1j}, & j \in S_1 \\ 0, & \text{else} \end{array} \right. \quad \overline{r}_{1j}(\mathbf{S}) = \left\{ \begin{array}{ll} r_{1j}, & j \in S_1 \\ 0, & \text{else} \end{array} \right. \tag{4}$$

so that the output of the $j$-th neuron at the first layer is $\overline{W}_{1j}(\mathbf{S})^T x + \overline{r}_{1j}(\mathbf{S})$. We recursively define $\overline{W}_{ij}(\mathbf{S})$ and $\overline{r}_{ij}(\mathbf{S})$ by letting $\overline{\mathbf{W}}_i(\mathbf{S})$ be a matrix with columns $\overline{W}_{i1}(\mathbf{S}), \ldots, \overline{W}_{iM_i}(\mathbf{S})$ and

$$\overline{W}_{ij}(\mathbf{S}) = \left\{ \begin{array}{ll} \overline{\mathbf{W}}_{i-1}(\mathbf{S})W_{ij}, & j \in S_i \\ 0, & \text{else} \end{array} \right. \quad \overline{r}_{ij}(\mathbf{S}) = \left\{ \begin{array}{ll} W_{ij}^T \overline{r}_{i-1}(\mathbf{S}) + r_{ij}, & j \in S_i, \\ 0, & \text{else} \end{array} \right. \tag{5}$$

where $\overline{\mathbf{r}}_i(\mathbf{S})$ is the vector with elements $\overline{r}_{ij}(\mathbf{S})$, $j = 1, \ldots, M_i$.

We define $\overline{W}(\mathbf{S}) = \overline{\mathbf{W}}_L(\mathbf{S})\Omega$ and $\overline{r}(\mathbf{S}) = \Omega^T \overline{\mathbf{r}}_L(\mathbf{S}) + \psi$. Based on these notations, when an input $x$ activates the set $\mathbf{S}$, $z_{ij} = \overline{W}_{ij}(\mathbf{S})^T x + \overline{r}_{ij}(\mathbf{S})$ and $y = \overline{W}(\mathbf{S})^T x + \overline{r}(\mathbf{S})$.

**Lemma 1.** *Let $\overline{\mathcal{X}}(\mathbf{S})$ denote the set of inputs $x$ that activate a particular set of neurons $\mathbf{S}$, and let $\overline{\mathbf{W}}_0(\mathbf{S})$ be equal to the identity matrix, and $\overline{\mathbf{r}}_0$ to be zero vector. Then*

$$\overline{\mathcal{X}}(\mathbf{S}) = \bigcap_{i=1}^{L} \left( \bigcap_{j \in S_i} \{x : W_{ij}^T(\overline{\mathbf{W}}_{i-1}(\mathbf{S})^T x + \overline{\mathbf{r}}_{i-1}) + r_{ij} \geq 0\} \right.$$

$$\left. \cap \bigcap_{j \notin S_i} \{x : W_{ij}^T(\overline{\mathbf{W}}_{i-1}(\mathbf{S})^T x + \overline{\mathbf{r}}_{i-1}) + r_{ij} \leq 0\} \right). \tag{6}$$

A proof can be found in the supplementary material. Note that in (6), a particular input $x$ could belong to multiple activation regions $\overline{\mathcal{X}}(\mathbf{S})$. This is because if the $j$-th neuron at the $i$-th layer is unstable, then both $(S_1, \ldots, S_i \cup \{j\}, \ldots, S_L)$ and $(S_1, \ldots, S_i \setminus \{j\}, \ldots, S_L)$ can be regarded as activated by $x$. We let $\mathbf{S}(x) \triangleq \{\mathbf{S} : x \in \overline{\mathcal{X}}(\mathbf{S})\}$ and let $\mathbf{T}(x)$ denote the set of unstable neurons produced by input $x$. Given a collection of activation sets $\mathbf{S}_1, \ldots, \mathbf{S}_r$, we let

$$\mathbf{T}(\mathbf{S}_1, \ldots, \mathbf{S}_r) = \left( \bigcup_{l=1}^{r} \mathbf{S}_l \right) \setminus \left( \bigcap_{l=1}^{r} \mathbf{S}_l \right).$$

The set $\mathbf{T}(\mathbf{S}_1, \ldots, \mathbf{S}_r)$ is equal to the set of neurons that must be unstable in order for an input $x$ to belong to $\overline{\mathcal{X}}(\mathbf{S}_1) \cap \ldots \cap \overline{\mathcal{X}}(\mathbf{S}_r)$.

## 3 Problem Formulation and Safety Conditions

In this section, we first formally define the problem, and then give necessary and sufficient conditions for verifying NCBFs.

### 3.1 Problem Formulation

We consider a feedforward neural network (NN) $b : \mathcal{X} \to \mathbb{R}$ with ReLU activation function. Since $b$ is not continuously differentiable due to the piecewise linearity of the ReLU function, the guarantees of Theorem 1 cannot be applied directly. Our overall goal will be to derive analogous conditions to (2) for ReLU-NCBFs, and then ensure that any control policy $\mu$ that satisfies the conditions will be safe with $\mathcal{D} = \{x : b(x) \geq 0\}$.

**Problem 1.** *Given a nonlinear continuous-time system (1), a neural network function $b : \mathcal{X} \to \mathbb{R}$, a set of admissible control inputs $\mathcal{U} = \{u : Au \leq c\}$ for given matrix $A \in \mathbb{R}^{p \times m}$ and vector $c \in \mathbb{R}^p$, and a safe set $\mathcal{C} = \{x : h(x) \geq 0\}$, determine whether (i) $\mathcal{D} \subseteq \mathcal{C}$ and (ii) there exists a control policy $\mu$ such that $\mathcal{D}$ is positive invariant under dynamics (1) and control policy $\mu$.*

## 3.2 Exact Conditions for Safety

When a CBF $b(x)$ is continuously differentiable, ensuring that (2) is satisfied is equivalent to verifying that there is no $x$ satisfying $b(x) = 0$, $\frac{\partial b}{\partial x} g(x) = 0$, and $\frac{\partial b}{\partial x} f(x) < 0$ [17]. When $b$ is represented by a ReLU neural network, however, $b$ will not be differentiable when the input $x$ leads to neurons having zero pre-activation input, i.e., when $\mathbf{T}(x) \neq \emptyset$. Although the set of $x$ with $\mathbf{T}(x) \neq \emptyset$ has measure zero, such points can nonetheless cause safety violations as illustrated by the following example.

**Example:** Let $x(t) \in \mathbb{R}^2$ and suppose the dynamics of $x$ are given by

$$\begin{array}{rcl} \dot{x}_1(t) & = & x_1 + u \\ \dot{x}_2(t) & = & -x_1 + 5x_2 \end{array} \quad \Rightarrow \quad f(x) = \begin{pmatrix} 1 & 0 \\ -1 & 5 \end{pmatrix} x, \; g(x) = \begin{pmatrix} 1 \\ 0 \end{pmatrix}$$

Suppose that $\mathcal{U} = \mathbb{R}^m$, the safe region $\mathcal{C} = \{x : x_1^2 + x_2^2 \leq 9\}$, and the candidate CBF is given by $b(x) = 1 - ||x||_1$. This CBF can be realized by a neural network with a single layer ($L = 1$) with four neurons ($M_1 = 4$), weights $W_{11} = (1\ 0)^T$, $W_{12} = (-1\ 0)^T$, $W_{13} = (0\ 1)^T$, $W_{14} = (0\ -1)^T$, $r_{1j} = 0$ for $j = 1, \ldots, 4$, $\Omega_j = -1$ for $j = 1, \ldots, 4$, and $\psi = 1$. We observe that $\mathcal{D} = \{x : b(x) \geq 0\} \subseteq \mathcal{C}$. Furthermore, whenever $\frac{\partial b}{\partial x}$ exists, we have $\frac{\partial b}{\partial x} \in \{(1\ 1), (1\ -1), (-1\ 1), (-1\ -1)\}$, each of which satisfies $\frac{\partial b}{\partial x} g(x) \neq 0$. On the other hand, the set $\mathcal{D}$ is not positive invariant. If $x_2(0) > \frac{1}{5}$, then $|x_1(0)| \leq 1$ implies that $\dot{x}_2(0) > 0$, and indeed, $x_2(t)$ will continue to increase until $x(t) \notin \mathcal{D}$. More details about this example can be found in Section A.17.

Fundamentally, this safety violation occurs because at $x = (0\ 1)^T$, we have $b(x) = 0$, $\mathbf{T}(x) = \{(1, 1), (1, 2)\}$ (i.e., the preactivation input to the first and second neurons in the hidden layer is zero), creating a discontinuity in the slope of $b(x)$. For $x'$ in the neighborhood of $(0\ 1)^T$, the value of $\frac{\partial b}{\partial x}(x')g(x')$ will be either 1 or $-1$. Since there is no single control input that satisfies (2) for both values of $\frac{\partial b}{\partial x} g(x')$, it is impossible to ensure safety of the system in the neighborhood of $(0\ 1)^T$.

The following lemma addresses this issue by giving exact and general conditions for a NCBF with ReLU activation function to satisfy positive invariance. We let $\partial \mathcal{D}$ denote the boundary of the set $\mathcal{D}$.

**Lemma 2.** *The set $\mathcal{D}$ is positive invariant under control policy $\mu$ if and only if, for all $x \in \partial \mathcal{D}$, there exist $\mathbf{S} \in \mathbf{S}(x)$ such that*

$$(\overline{\mathbf{W}}_{i-1}(\mathbf{S})W_{ij})^T(f(x) + g(x)\mu(x)) \geq 0 \quad \forall (i, j) \in \mathbf{T}(x) \cap \mathbf{S} \tag{7}$$

$$(\overline{\mathbf{W}}_{i-1}(\mathbf{S})W_{ij})^T(f(x) + g(x)\mu(x)) \leq 0 \quad \forall (i, j) \in \mathbf{T}(x) \setminus \mathbf{S} \tag{8}$$

$$\overline{W}(\mathbf{S})^T(f(x) + g(x)\mu(x)) \geq 0 \tag{9}$$

The proof is based on the generalized Nagumo's Theorem [37] and a novel characterization of the Clarke tangent cone to $\partial \mathcal{D}$, and can be found in the supplementary material due to space constraints. Intuitively, conditions (7)–(9) can be interpreted as follows. Condition (9) is similar to (2) when the gradient is given by $\overline{W}(\mathbf{S})$, i.e., when $x$ is in the interior of the activation region defined by $\mathbf{S}$. Eq. (7)–(8) can be interpreted as choosing the control input to ensure that $x$ remains in the activation region of $\mathbf{S}$ (i.e., $\overline{\mathcal{X}}(\mathbf{S})$).

Lemma 2 can be used to construct NCBF-based safe control policies, analogous to control policies based on continuously differentiable CBFs. Formally, given a nominal control policy $\mu_{nom}$, at each time $t$, we can choose $u(t)$ by solving the optimization problem

$$\begin{array}{ll} \min_{\mathbf{S} \in \mathbf{S}(x), u} & ||u - \mu_{nom}(x(t))||_2^2 \\ \text{s.t.} & \overline{W}(\mathbf{S})^T(f(x) + g(x)\mu(x)) \geq -\alpha(b(x)) \\ & u \in \mathcal{U}, (7), (8) \end{array} \tag{10}$$

where $\alpha$ satisfies the same conditions as in Theorem 1. This problem can be solved by decomposing (10) into $|\mathbf{S}(x)|$ quadratic programs (one for each $\mathbf{S}$) and then selecting the value of $u$ that minimizes the objective function across all of the QPs. In particular, if $\mathbf{S}(x)$ is a singleton, then the constraints on (10) reduce to Eq. (2).

We then give an equivalent condition for $b(x)$ to be an NCBF. As a preliminary, we say that a collection of activation sets $\mathbf{S}_1, \ldots, \mathbf{S}_r$ is *complete* if for any $\mathbf{S}' \notin \{\mathbf{S}_1, \ldots, \mathbf{S}_r\}$, we have $\overline{\mathcal{X}}(\mathbf{S}_1) \cap \cdots \cap \overline{\mathcal{X}}(\mathbf{S}_r) \cap \overline{\mathcal{X}}(\mathbf{S}') = \emptyset$.

**Proposition 1.** *The function $b$ is a valid CBF iff the following two properties hold:*

*(i) For all activation sets $\mathbf{S}_1, \ldots, \mathbf{S}_r$ with $\{\mathbf{S}_1, \ldots, \mathbf{S}_r\}$ complete and any $x$ satisfying $b(x) = 0$ and*

$$x \in \left( \bigcap_{l=1}^{r} \overline{\mathcal{X}}(\mathbf{S}_l) \right), \tag{11}$$

*there exist $l \in \{1, \ldots, r\}$ and $u \in \mathcal{U}$ such that*

$$(\overline{\mathbf{W}}_{i-1}(\mathbf{S}_l) W_{ij})^T (f(x) + g(x)u) \geq 0 \quad \forall (i,j) \in \mathbf{T}(\mathbf{S}_1, \ldots, \mathbf{S}_r) \cap \mathbf{S}_l \tag{12}$$

$$(\overline{\mathbf{W}}_{i-1}(\mathbf{S}_l) W_{ij})^T (f(x) + g(x)u) \leq 0 \quad \forall (i,j) \in \mathbf{T}(\mathbf{S}_1, \ldots, \mathbf{S}_r) \setminus \mathbf{S}_l \tag{13}$$

$$\overline{W}(\mathbf{S}_l)^T (f(x) + g(x)u) \geq 0 \tag{14}$$

*(ii) For all activation sets $\mathbf{S}$, we have*

$$(\overline{\mathcal{X}}(\mathbf{S}) \cap \mathcal{D}) \setminus \mathcal{C} = \emptyset \tag{15}$$

The proof can be found in the supplementary material. We refer to any $x$ that fails to meet at least one of conditions (i) and (ii) of Proposition 1 as a safety counterexample.

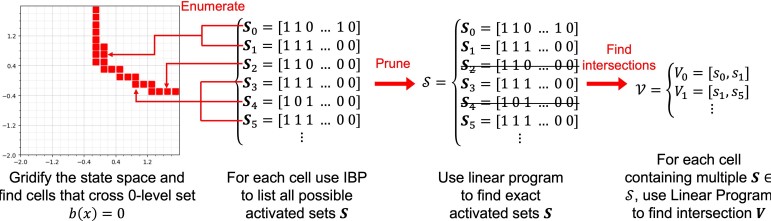

Figure 1: Illustration of proposed coarser-to-finer searching method. Hyper-cubes that intersect the safety boundaries are marked in red. When all possible activation sets are listed, we can identify exact activation set and intersections.

## 4 Verification Algorithm

The preceding proposition motivates our overall approach to verifying NCBFs, consisting of the following steps. **(Step 1)** We conduct coarser-to-finer search by discretizing the state space into hyper-cubes and use linear relaxation based perturbation analysis (LiRPA) to identify grid squares that intersect the boundary $\{x : b(x) = 0\}$. **(Step 2)** We enumerate all possible activation sets within each hyper-cube using Interval Bound Propagation (IBP). We then identify the activation sets and intersections that satisfy $b(x) = 0$ using linear programming. **(Step 3)** For each activation set and intersection of activation sets, we verify the conditions of Proposition 1. In what follows, we describe each step in detail.

### 4.1 Enumeration of Activation Set Boundaries

We next present the approach to enumerate activation sets of a given ReLU NCBF that intersect the safety boundary, $b(x) = 0$. Our approach first conducts a coarse-grained search that over-approximates the collection of activation sets that intersect with the safety boundary. We then prune this collection to remove activation sets that cannot be realized. Specifically, we first identify regions that contain $b(x) = 0$, then enumerate all possible activation sets within the region and finally identify the sets that intersect the safety boundaries.

We first discretize the given state space $\mathcal{X}$ into hyper-cubes. We compute lower and upper bounds of $b(x)$ on each hyper-cube, denoted $b_l$ and $b_u$ respectively, with linear relaxation based perturbation analysis (LiRPA), i.e., auto LiRPA [38]. For each hyper-cube $B$, we determine $B$ contains $b(x) = 0$ (red squares in Fig 1) if criteria $sgn(b_l) \cdot sgn(b_u) \leq 0$ holds, where $sgn(*)$ is the sign function. For each $B$ contains $b(x) = 0$, we utilize IBP to over-approximate unstable neurons. Details on this IBP procedure can be found in Section A.7. Let $\tilde{\mathcal{S}}$ denote the over-approximated collection of activation

sets computed by IBP. Note that $S \in \tilde{\mathcal{S}}$ may not intersect with $b(x) = 0$. As shown in Fig. 1, we let $\mathcal{S}$ denote the collection of activation sets $\mathbf{S} \in \tilde{\mathcal{S}}$ that satisfy

$$\overline{W}(\mathbf{S})^T x + \overline{r}(\mathbf{S}) = 0$$
$$(\overline{\mathbf{W}}_{i-1}(\mathbf{S})W_{ij})^T x + \overline{r}_{ij}(\mathbf{S}) \geq 0 \ \forall (i,j) \in \mathbf{S} \tag{16}$$
$$(\overline{\mathbf{W}}_{i-1}(\mathbf{S})W_{ij})^T x + \overline{r}_{ij}(\mathbf{S}) \leq 0 \ \forall (i,j) \notin \mathbf{S}$$

for some $x \in B$. The activation set boundaries indicate unstable neurons, which need to be verified separately. Hence, we search for intersections as follows, where $2^{\mathcal{S}}$ denotes the set of subsets of $\mathcal{S}$.

$$\mathcal{V} = \left\{ Z \in 2^{\mathcal{S}} : \left( \bigcap_{\mathbf{S} \in Z} \overline{\mathcal{X}}(\mathbf{S}) \right) \cap \{x : b(x) = 0\} \neq \emptyset \right\}$$

## 4.2 Verifying Safety of Each Activation Set

With the activation sets enumerated, the following result gives an equivalent condition for verifying that there is no safety counterexample in the interior of $\overline{\mathcal{X}}(\mathbf{S})$ for a given activation set $\mathbf{S}$.

**Lemma 3.** *There is no safety counterexample in the set $\overline{\mathcal{X}}(\mathbf{S}) \setminus \bigcup_{\mathbf{S}' \neq \mathbf{S}} \overline{\mathcal{X}}(\mathbf{S})$ if and only if:*

1. *There do not exist $x \in \mathcal{X}$ and $y \in \mathbb{R}^{p+1}$ satisfying (a) $x \in int(\overline{\mathcal{X}}(\mathbf{S}))$, (b) $b(x) = 0$, (c) $y \geq 0$, (d) $y^T \begin{pmatrix} -\overline{W}(\mathbf{S})^T g(x) \\ A \end{pmatrix} = 0$, and (e) $y^T \begin{pmatrix} \overline{W}(\mathbf{S})^T f(x) \\ c \end{pmatrix} < 0$.*

2. *There does not exist $x \in \mathcal{X}$ with $b(x) = 0$, $x \in int(\overline{\mathcal{X}}(\mathbf{S}))$, and $h(x) < 0$.*

The proof can be found in the supplementary material. Conditions 1 and 2 of Lemma 3 can be verified by solving nonlinear programs (see supplementary material for the formulations of these nonlinear programs). Such nonlinear programs are solved for each activation set $\mathbf{S}$ in the collection $\mathcal{S}$ computed during the enumeration phase. The following corollary describes the special case where there are no constraints on the control, i.e., $\mathcal{U} = \mathbb{R}^m$.

**Corollary 1.** *If $\mathcal{U} = \mathbb{R}^m$, then there is no safety counterexample in the set $\overline{\mathcal{X}}(\mathbf{S}) \setminus \bigcup_{\mathbf{S}' \neq \mathbf{S}} \overline{\mathcal{X}}(\mathbf{S}')$ if and only if (1) there does not exist $x \in \mathcal{X}$ satisfying $x \in int(\overline{\mathcal{X}}(\mathbf{S}))$, $b(x) = 0$, $\overline{W}(\mathbf{S})^T g(x) = 0$, and $\overline{W}(\mathbf{S})^T f(x) < 0$ and (2) there does not exist $x \in \mathcal{X}$ with $b(x) = 0$, $x \in int(\overline{\mathcal{X}}(\mathbf{S}))$, and $h(x) < 0$.*

Under the conditions of Corollary 1, the complexity of the verification problem can be reduced. If $g(x)$ is a constant matrix $G$, then safety is automatically guaranteed if $\overline{W}(\mathbf{S})^T G \neq 0$. If $f(x)$ is linear in $x$ as well, then verification can be performed via a linear program.

## 4.3 Verification of Activation Set Intersections

With the activation set boundaries enumerated, we now describe an approach for verifying safety of intersections between activation sets.

**Lemma 4.** *Suppose that the sets $\mathbf{S}_1, \ldots, \mathbf{S}_r$ satisfy condition (ii) of Lemma 3. There is no safety counterexample in the set*

$$\overline{\mathcal{X}}(\mathbf{S}_1) \cap \cdots \cap \overline{\mathcal{X}}(\mathbf{S}_r) \setminus \bigcup_{S' \notin \{\mathbf{S}_1, \ldots, \mathbf{S}_r\}} \overline{\mathcal{X}}(\mathbf{S}')$$

*if and only if there do not exist $x \in \mathcal{X}$ and $y_1, \ldots, y_r \in \mathbb{R}^{T+p+1}$, where $T = |\mathbf{T}(\mathbf{S}_1, \ldots, \mathbf{S}_r)|$, satisfying (a) $x \in int(\overline{\mathcal{X}}(\mathbf{S}_1)) \cap \cdots \cap int(\overline{\mathcal{X}}(\mathbf{S}_r))$, (b) $b(x) = 0$, (c) $y_l \geq 0 \ \forall l$, (d) $\forall l = 1, \ldots, r$, $y_l^T \Theta_l(\mathbf{S}_1, \ldots, \mathbf{S}_r(x)) = 0$, where $\Theta_l(\mathbf{S}_1, \ldots, \mathbf{S}_r, x)$ is a $(T + p + 1) \times m$ matrix*

$$\Theta_l(\mathbf{S}_1, \ldots, \mathbf{S}_r, x) \triangleq \begin{pmatrix} -(\overline{\mathbf{W}}_{i-1}(\mathbf{S}_l)W_{ij})^T g(x) : (i,j) \in \mathbf{T}(\mathbf{S}_1, \ldots, \mathbf{S}_r) \cap \mathbf{S}_l \\ (\overline{\mathbf{W}}_{i-1}(\mathbf{S}_l)W_{ij})^T g(x) : (i,j) \in \mathbf{T}(\mathbf{S}_1, \ldots, \mathbf{S}_r) \setminus \mathbf{S}_l \\ -W(\mathbf{S}_l)^T g(x) \\ A \end{pmatrix} \tag{17}$$

*and (e)* $\forall l = 1, \ldots, r$, $y_l^T \Lambda_l(\mathbf{S}_1, \ldots, \mathbf{S}_r, x) < 0$, *where* $\Lambda_l(\mathbf{S}_1, \ldots, \mathbf{S}_r, x) \in \mathbb{R}^{T+p+1}$ *is given by*

$$\Lambda_l(\mathbf{S}_1, \ldots, \mathbf{S}_r, x) \triangleq \begin{pmatrix} (\overline{\mathbf{W}}_{i-1}(\mathbf{S}_l) W_{ij})^T f(x) : (i, j) \in \mathbf{T}(\mathbf{S}_1, \ldots, \mathbf{S}_r) \cap \mathbf{S}_l \\ -(\overline{\mathbf{W}}_{i-1}(\mathbf{S}_l) W_{ij})^T f(x) : (i, j) \in \mathbf{T}(\mathbf{S}_1, \ldots, \mathbf{S}_r) \setminus \mathbf{S}_l \\ \overline{W}(\mathbf{S}_l)^T f(x) \\ c \end{pmatrix} \quad (18)$$

The proof can be found in the supplementary material. The verification problem can be mapped to solving a nonlinear program. This nonlinear program is solved for each $(\mathbf{S}_1, \ldots, \mathbf{S}_r) \in \mathcal{V}$, where $\mathcal{V}$ is computed during the enumeration phase. Note that, if $g(x)$ is constant, then the constraints of the nonlinear program are linear, and if $f(x)$ is linear then the problem reduces to solving a system of linear and bilinear inequalities. Furthermore, if $f(x)$ can be bounded over $x \in \overline{\mathcal{X}}(\mathbf{S}_1) \cap \cdots \cap \overline{\mathcal{X}}(\mathbf{S}_r)$ and $b(x) = 0$, then this bound can be used to derive a linear relaxation to the objective function of (27), thus relaxing the nonlinear program to a linear program.

The complexity of this approach can also be reduced by utilizing sufficient conditions for safety that are easier to check. For instance, if there exists $u \in \mathcal{U}$ such that $\overline{W}(\mathbf{S})^T(f(x) + g(x)u) \geq 0$ for all $\mathbf{S} \in \mathbf{S}(x)$, then the criteria of Lemma 4 are satisfied. In particular, if $\overline{W}(\mathbf{S})^T f(x) \geq 0$ for all $\mathbf{S} \in \mathbf{S}(x)$, then the conditions are satisfied with $u = 0$. The following result is a straightforward consequence of Proposition 1, Lemma 3, and Lemma 4.

**Theorem 2.** *Suppose that the conditions of Lemma 3 are satisfied for each* $\mathbf{S} \in \mathcal{S}$ *and the conditions of Lemma 4 are satisfied for each* $\{\mathbf{S}_1, \ldots, \mathbf{S}_r\} \in \mathcal{V}$. *Then the NCBF* $b$ *satisfies the conditions of Problem 1.*

## 5 Experiments

In this section, we evaluate our proposed method to verify neural control barrier functions on three systems, namely Darboux, obstacle avoidance and spacecraft rendezvous. The experiments run on a 64-bit Windows PC with Intel i7-12700 processor, 32GB RAM and NVIDIA GeForce RTX 3080 GPU. We include experiment details in the supplement.

### 5.1 Experiment Setup

**Darboux:** We consider the Darboux system [39], a nonlinear open-loop polynomial system that has been widely used as a benchmark for constructing barrier certificates. The dynamic model is given in the supplement. We obtain the trained NCBF by following the method proposed in [24].

**Obstacle Avoidance (OA):** We evaluate our proposed method on a controlled system [40]. We consider an Unmanned Aerial Vehicles (UAVs) avoiding collision with a tree trunk. We model the system as a Dubins-style [41] aircraft model. The system state consists of 2-D position and aircraft yaw rate $x := [x_1, x_2, \psi]^T$. We let $u$ denote the control input to manipulate yaw rate and the dynamics defined in the supplement. We train the NCBF via the method proposed in [24] with $v$ assumed to be 1 and the control law $u$ designed as $u = \mu_{nom}(x) = -\sin \psi + 3 \cdot \frac{x_1 \cdot \sin \psi + x_2 \cdot \cos \psi}{0.5 + x_1^2 + x_2^2}$.

**Spacecraft Rendezvous (SR):** We evaluate our approach on a spacecraft rendezvous problem from [42]. A station-keeping controller is required to keep the "chaser" satellite within a certain relative distance to the "target" satellite. The state of the chaser is expressed relative to the target using linearized Clohessy–Wiltshire–Hill equations, with state $x = [p_x, p_y, p_z, v_x, v_y, v_z]^T$, control input $u = [u_x, u_y, u_z]^T$ and dynamics defined in the supplement. We train the NCBF as in [13].

**hi-ord$_8$:** We evaluate our approach on an eight-dimensional system that first appeared in [21] to evaluate the scalability of proposed verification method.

### 5.2 Experiment Results

We first verify that $\mathcal{D} \subseteq \mathcal{C}$ in the Darboux and obstacle avoidance scenarios. We trained four 1 hidden layer NCBFs with 20, 5, 32, 10 neurons, respectively. NCBF (a) and (c) completed training, (b) terminated after 3 epochs and (d) terminated after 5 epochs. Our verification algorithm identifies safety violations. NCBF (a) and (c) pass the verification while NCBF (b) and (d) fail. As shown in

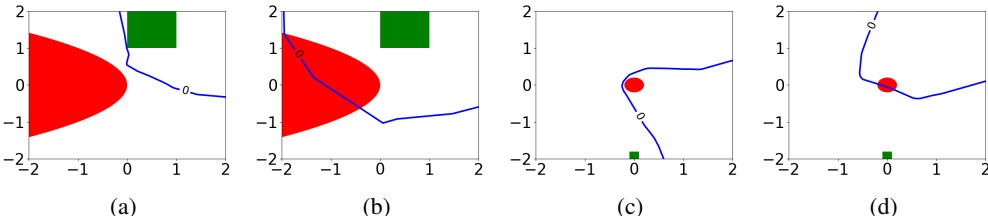

$$\begin{array}{cccc} \text{(a)} & \text{(b)} & \text{(c)} & \text{(d)} \end{array}$$

Figure 2: Comparison of NCBFs that pass and fail the proposed verification. We show 0-level set boundary in blue, initial region in green and the unsafe region in red. (a) and (b) visualize NCBFs for Darboux. (c) and (d) shows projection of NCBFs for Obstacle Avoidance.

Fig. 2a, the fully-trained NCBF (a) separates the safe and unsafe regions while the NCBF (b) with unfinished training has a boundary that intersects the unsafe region in Fig. 2b. As shown in Fig. 2d, the 2-D projection with $\psi = -0.5$ of NCBF (d) shows its boundary overlap the unsafe region.

We next verify the positive invariance of $\mathcal{D}$. All NCBFs for Darboux pass feasibility verification with run-time presented in Table 1. We compare the run-time of proposed verification with SMT-based approaches using translator proposed in [21] and SMT solver dReal [43] and Z3 [44]. The proposed method can verify NCBFs with more ReLU neurons while SMT-based verifier return nothing. All of

Table 1: Comparison of verification run-time of NCBF in seconds. The table contains the dimension $n$, network architecture with $\sigma$ denoting ReLU, the number of activation sets $N$ and run-time of proposed method including time of enumerating, verification and total run-time denoted as $t_e$, $t_v$ and $T$, respectively. We compare with the run-time of dReal ($T_{dReal}$) and Z3 ($T_{Z3}$).

| Case | n | NN Architecture | N | $t_e$ | $t_v$ | T | $T_{dReal}$ | $T_{Z3}$ |
|------|---|-----------------|---|-------|-------|---|-------------|----------|
| Darboux | 2 | 2-20-$\sigma$-1 | 13 | 0.39 | 0.04 | 0.43 | 0.36 | >3hrs |
| | 2 | 2-32-$\sigma$-1 | 20 | 0.23 | 0.03 | 0.27 | 1.27 | >3hrs |
| | 2 | 2-64-$\sigma$-1 | 47 | 1.89 | 0.13 | 2.02 | >3hrs | >3hrs |
| | 2 | 2-96-$\sigma$-1 | 63 | 4.17 | 0.14 | 4.31 | >3hrs | >3hrs |
| | 2 | 2-128-$\sigma$-1 | 65 | 4.77 | 0.03 | 4.90 | >3hrs | >3hrs |
| | 2 | 2-512-$\sigma$-1 | 258 | 33.44 | 0.61 | 34.05 | >3hrs | >3hrs |
| | 2 | 2-1024-$\sigma$-1 | 548 | 107.63 | 0.81 | 108.44 | >3hrs | >3hrs |
| Darboux | 2 | 2-10-$\sigma$-10-$\sigma$-1 | 4 | 0.42 | 0.05 | 0.47 | 7.70 | 15.81 |
| | 2 | 2-16-$\sigma$-16-$\sigma$-1 | 26 | 2.13 | 0.11 | 2.23 | >3hrs | >3hrs |
| | 2 | 2-32-$\sigma$-32-$\sigma$-1 | 58 | 8.35 | 6.28 | 14.64 | >3hrs | >3hrs |
| | 2 | 2-48-$\sigma$-48-$\sigma$-1 | 58 | 13.69 | 0.19 | 13.88 | >3hrs | >3hrs |
| | 2 | 2-64-$\sigma$-64-$\sigma$-1 | 102 | 31.30 | 0.47 | 31.77 | >3hrs | >3hrs |
| | 2 | 2-256-$\sigma$-256-$\sigma$-1 | 402 | 319.81 | 1.92 | 321.73 | >3hrs | >3hrs |
| | 2 | 2-512-$\sigma$-512-$\sigma$-1 | 1150 | 619.10 | 0.75 | 619.85 | >3hrs | >3hrs |
| hi-ord$_8$ | 8 | 8-8-$\sigma$-1 | 98 | 3.70 | 0.02 | 3.72 | >3hrs | >3hrs |
| | 8 | 8-16-$\sigma$-1 | 37 | 35.05 | 0.0 | 35.05 | >3hrs | >3hrs |

the generated NCBFs for obstacle avoidance and spacecraft rendezvous passed the verification. The run-times are presented in Table 2. We find that the activation set sizes grow with the size of neural network and the state dimension $n$. We conjecture that this growth rate could be reduced by using tighter approximations for the activated sets.

A key feature of our approach is that the verification process does not depend on the control policy $\mu$, but rather we verify that any $\mu$ satisfying (7)–(9) for all $x \in \partial \mathcal{D}$ is safe. This improves flexibility while reducing false negatives in safety verification, as we illustrate in the following example. The NCBF for obstacle avoidance with one hidden layer and 32 neurons fails the SMT-based verification using dReal given nominal controller $\mu_{nom}$. The counter example is at the point $(-0.20, 0, -0.73)$. At this point, we have $b(x) = 0.0033$, $\frac{\partial b}{\partial x} f(x) = 0.0095$, which satisfies the sufficient conditions for safety in Lemma 2. However, the nominal controller yields $\frac{\partial b}{\partial x}(f(x) + g(x)u) = -1.30$ and hence

fails verification. Hence, by following a safe control policy of the form (10) with the learned value of $b$ and nominal policy $\mu_{nom}$, we can retain the performance of $\mu_{nom}$ while still ensuring safety.

Table 2: Comparison of verification run-time of NCBF in seconds. We denote the run-time as 'UTD' when the method is unable to be directly used for verification.

| Case | n | NN Architecture | $N$ | $t_e$ | $t_v$ | $T$ | $T_{dReal}$ | $T_{Z3}$ |
|---|---|---|---|---|---|---|---|---|
| OA | 3 | 3-32-$\sigma$-1 | 436 | 6.94 | 0.40 | 7.35 | >6hrs | >6hrs |
| | 3 | 3-64-$\sigma$-1 | 1645 | 19.17 | 1.87 | 21.05 | >6hrs | >6hrs |
| | 3 | 3-96-$\sigma$-1 | 3943 | 58.96 | 6.26 | 65.22 | >6hrs | >6hrs |
| | 3 | 3-128-$\sigma$-1 | 5695 | 169.51 | 15.56 | 185.07 | >6hrs | >6hrs |
| OA | 3 | 3-16-$\sigma$-16-$\sigma$-1 | 467 | 18.72 | 2.66 | 21.39 | >6hrs | >6hrs |
| | 3 | 3-32-$\sigma$-32-$\sigma$-1 | 1324 | 218.86 | 54.51 | 273.37 | >6hrs | >6hrs |
| | 3 | 3-48-$\sigma$-48-$\sigma$-1 | 3754 | 3197.59 | 9.75 | 3207.34 | >6hrs | >6hrs |
| | 3 | 3-64-$\sigma$-64-$\sigma$-1 | 6545 | 11730.28 | 18.22 | 11748.50 | >6hrs | >6hrs |
| SR | 6 | 6-8-$\sigma$-8-$\sigma$-8-1 | 270 | 78.77 | 10.23 | 89.00 | UTD | UTD |
| | 6 | 6-16-$\sigma$-16-$\sigma$-16-1 | 32937 | 1261.60 | 501.06 | 1762.67 | UTD | UTD |
| | 6 | 6-32-$\sigma$-32-$\sigma$-32-1 | 207405 | 12108.11 | 1798.08 | 13906.19 | UTD | UTD |

The computational complexity of our approach will be determined by several factors including the dimension of the state, the number of layers, the number of neurons in each layer, and the geometry of the 0-level set of the NCBF. As shown in Table 2, the dimension of the system plays the most important role.

## 6   Conclusion

This paper studied the problem of verifying safety of a nonlinear control system using a NCBF represented by a feed-forward neural network with ReLU activation function. We leveraged a generalization of Nagumo's theorem for proving invariance of sets with nonsmooth boundaries to derive necessary and sufficient conditions for safety. The exact safety conditions addressed the issue that ReLU NCBFs will be nondifferentiable at certain points, thus invalidating traditional safety verification methods. Based on this condition, we proposed an algorithm for safety verification of NCBFs that first decomposes the NCBF into piecewise linear segments and then solves a nonlinear program to verify safety of each segment as well as the intersections of the linear segments. We mitigated the complexity by only considering the boundary of the safe region and pruning the segments with IBP and LiRPA. We evaluated our approach through numerical studies with comparison to state-of-the-art SMT-based methods. Future extensions of this work could include improving scalability for higher-dimensional systems and deeper neural networks, as well as other activation functions.

## Acknowledgements

This research was partially supported by the NSF (grants CNS-1941670, ECCS-2020289, IIS-1905558, and IIS-2214141), AFOSR (grants FA9550-22-1-0054 and FA9550-23-1-0208), and ARO (grant W911NF-19-1-0241).

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

# A  Supplementary Material

In what follows, we give some details of content omitted in the paper due to space limit. The supplements are organized as follows. We give some proof of Lemma 1, 2, Proposition 1, Lemma 3, 4, and Theorem 2 in Section A.1 –A.6, respectively. We present system dynamics for Darboux, obstacle avoidance, spacecraft rendezvous and hi-ord$_8$ in Section A.9–A.12. We provide some training details in Section A.13 as well as experiment details and results in Section A.14. We compare polynomial CBFs with NCBF in A.15, compare NCBFs with different activation functions in A.16. We present more details in A.17 on the example in 3.2.

## A.1  Proof of Lemma 1

We prove by induction on $L$. If $L = 1$, then $x \in \overline{\mathcal{X}}(\mathbf{S})$ if the pre-activation input to the $(1, j)$ neuron is nonnegative for all $j \in S_1$ and nonpositive for all $j \notin S_1$. We have that the pre-activation input is equal to $W_{1j}^T x + r_{1j}$, establishing the result for $L = 1$.

Now, inducting on $L$, we have that $x \in \overline{\mathcal{X}}(S_1, \ldots, S_{L-1})$ if and only if

$$x \in \bigcap_{i=1}^{L-1} \left( \bigcap_{j \in S_i} \{x : W_{ij}^T(\overline{\mathbf{W}}_{i-1}(\mathbf{S})^T x + \bar{\mathbf{r}}_{i-1}) + r_{ij} \geq 0\} \right.$$

$$\left. \cap \bigcap_{j \notin S_i} \{x : W_{ij}^T(\overline{\mathbf{W}}_{i-1}(\mathbf{S})^T x + \bar{\mathbf{r}}_{i-1}) + r_{ij} \leq 0\} \right)$$

by induction. If $x \in \overline{\mathcal{X}}(S_1, \ldots, S_{L-1})$, then $x \in \overline{\mathcal{X}}(S_L)$ if and only if the pre-activation input to the $j$-th neuron at layer $L$ is nonnegative for all $j \in S_L$ and nonpositive for $j \notin S_L$. The pre-activation input is equal to $W_{Lj}^T z_{L-1} + r_{Lj}$, which we can expand by induction as

$$
\begin{aligned}
W_{Lj}^T z_{L-1} + r_{Lj} &= \sum_{j'=1}^{M_{L-1}} (W_{Lj})_{j'} z_{L-1,j'} + r_{Lj} \\
&= \sum_{j'=1}^{M_{L-1}} (W_{Lj})_{j'}(\overline{\mathbf{W}}_{L-1,j'}(\mathbf{S})^T x + \overline{\mathbf{r}_{L-1,j'}}(\mathbf{S}) + r_{Lj} \\
&= \left( \sum_{j'=1}^{M_{L-1}} (W_{Lj})_{j'} \overline{\mathbf{W}}_{L-1,j'}(\mathbf{S}) \right)^T x + \bar{\mathbf{r}}_{Lj}(\mathbf{S}) \\
&= (\overline{\mathbf{W}}_{L-1}(\mathbf{S}) W_{Lj})^T x + \bar{\mathbf{r}}_{Lj}(\mathbf{S})
\end{aligned}
$$

completing the proof.

## A.2  Proof of Lemma 2

The proof approach is based on Nagumo's Theorem, which gives necessary and sufficient conditions for positive invariance of a set. We first define the concept of tangent cone, and then present positive invariance conditions based on the tangent cone. The approach of the proof is to characterize the tangent cone to the set $\mathcal{D} = \{x : b(x) \geq 0\}$.

**Definition 2.** *Let $\mathcal{A}$ be a closed set. The tangent cone to $\mathcal{A}$ at $x$ is defined by*

$$\mathcal{T}_\mathcal{A}(x) = \left\{ z : \liminf_{\tau \to 0} \frac{dist(x + \tau z, \mathcal{A})}{\tau} = 0 \right\} \tag{19}$$

The following result gives an approach for constructing the tangent cone.

**Lemma 5** ([37])**.** *Suppose that the set $\mathcal{A}$ is defined by*

$$\mathcal{A} = \{x : q_k(x) \leq 0, k = 1, \ldots, N\}$$

*for some collection of differentiable functions $q_1, \ldots, q_N$. For any $x$, let $J(x) = \{k : q_k(x) = 0\}$. Then*

$$\mathcal{T}_\mathcal{A}(x) = \{z : z^T \nabla q_k(x) \le 0 \; \forall k \in J(x)\}.$$

The following is a fundamental preliminary result for establishing positive invariance.

**Theorem 3** (Nagumo's Theorem [37], Section 4.2). *A closed set $\mathcal{A}$ is controlled positive invariant if and only if, whenever $x(t) \in \partial\mathcal{A}$, $u(t) \in \mathcal{U}$ satisfies*

$$(f(x(t)) + g(x(t))u(t)) \in \mathcal{T}_\mathcal{A}(x(t)) \tag{20}$$

The following lemma characterizes the tangent cone to $\mathcal{D}$.

**Proposition 2.** *For any $x \in \partial\mathcal{D}$, we have*

$$\mathcal{T}_\mathcal{D}(x) = \bigcup_{\mathbf{S} \in \mathbf{S}(x)} \left[ \left( \bigcap_{(i,j) \in \mathbf{T}(x) \cap \mathbf{S}} \{z : (\overline{\mathbf{W}}_{i-1}(\mathbf{S})W_{ij})^T z \ge 0\} \right) \cap \right.$$
$$\left. \left( \bigcap_{(i,j) \in \mathbf{T}(x) \setminus \mathbf{S}} \{z : (\overline{\mathbf{W}}_{i-1}(\mathbf{S})W_{ij})^T z \le 0\} \right) \cap \{z : \overline{W}(\mathbf{S})^T z \ge 0\} \right] \tag{21}$$

*Proof.* Define $\overline{\mathcal{X}}_0(\mathbf{S}) = \overline{\mathcal{X}}(\mathbf{S}) \cap \mathcal{D}$. We will first show that, for all $x$ with $b(x) = 0$,

$$\mathcal{T}_\mathcal{D}(x) = \bigcup_{\mathbf{S} \in \mathbf{S}(\mathbf{x})} \mathcal{T}_{\overline{\mathcal{X}}_0(\mathbf{S})}(x). \tag{22}$$

We observe that

$$\mathrm{dist}\left( x, \mathcal{D} \setminus \bigcup_{\mathbf{S} \in \mathbf{S}(x)} \overline{\mathcal{X}}_0(\mathbf{S}) \right) > 0,$$

and hence

$$\mathrm{dist}(x + \tau z, \mathcal{D}) = \min_{\mathbf{S} \in \mathbf{S}(x)} \mathrm{dist}(x + \tau z, \overline{\mathcal{X}}_0(\mathbf{S}))$$

for $\tau$ sufficiently small.

Suppose that $z \in \mathcal{T}_{\overline{\mathcal{X}}_0(\mathbf{S})}(x)$. Then for any $\tau \ge 0$, $\mathrm{dist}(x + \tau z, \mathcal{D}) \le \mathrm{dist}(x + \tau z, \overline{\mathcal{X}}_0(\mathbf{S}))$ since $\overline{\mathcal{X}}_0(\mathbf{S}) \subseteq \mathcal{D}$, and hence

$$\liminf_{\tau \to 0} \frac{\mathrm{dist}(x + \tau z, \mathcal{D})}{\tau} \le \liminf_{\tau \to 0} \frac{\mathrm{dist}(x + \tau z, \overline{\mathcal{X}}_0(\mathbf{S}))}{\tau} = 0.$$

We therefore have $z \in \mathcal{T}_\mathcal{D}(x)$.

Now, suppose that $z \in \mathcal{T}_\mathcal{D}(\mathbf{x})$ and yet $z \notin \bigcup_{\mathbf{S} \in \mathbf{S}(x)} \mathcal{T}_{\overline{\mathcal{X}}_0(\mathbf{S})}(x)$. Then for all $\mathbf{S} \in \mathbf{S}(x)$, there exists $\epsilon_\mathbf{S} > 0$ such that

$$\liminf_{\tau \to 0} \frac{\mathrm{dist}(x + \tau z, \overline{\mathcal{X}}_0(\mathbf{S}))}{\tau} = \epsilon_\mathbf{S}.$$

Let $\overline{\epsilon} = \min\{\epsilon_\mathbf{S} : \mathbf{S} \in \mathbf{S}(x)\}$. For any $\delta \in (0, \overline{\epsilon})$, there exists $\overline{\tau} > 0$ such that $\tau < \overline{\tau}$ implies

$$\frac{\mathrm{dist}(x + \tau z, \mathcal{D})}{\tau} = \min_{\mathbf{S} \in \mathbf{S}(x)} \frac{\mathrm{dist}(x + \tau z, \overline{\mathcal{X}}_0(\mathbf{S}))}{\tau} > \delta$$

implying that $\liminf_{\tau \to 0} \frac{\mathrm{dist}(x + \tau z, \mathcal{D})}{\tau} > 0$ and hence $z \notin \mathcal{T}_\mathcal{D}(x)$. This contradiction implies (22).

It now suffices to show that, for each $\mathbf{S} \in \mathbf{S}(x)$,

$$\mathcal{T}_{\overline{\mathcal{X}}_0(\mathbf{S})}(x) = \left( \bigcap_{(i,j) \in \mathbf{T}(x) \cap \mathbf{S}} \{z : (\overline{\mathbf{W}}_{i-1}(\mathbf{S})W_{ij})^T z \ge 0\} \right) \cap$$
$$\left( \bigcap_{(i,j) \in \mathbf{T}(x) \setminus \mathbf{S}} \{z : (\overline{\mathbf{W}}_{i-1}(\mathbf{S})W_{ij})^T z \le 0\} \right) \cap \{z : \overline{W}(\mathbf{S})^T z \ge 0\}.$$

We have that each $\overline{\mathcal{X}}_0(\mathbf{S})$ is given by

$$\overline{\mathcal{X}}_0(\mathbf{S}) = \{x' : (\overline{\mathbf{W}}_{i-1}(\mathbf{S})W_{ij})^T x' + r_{ij}(\mathbf{S}) \geq 0 \ \forall (i,j) \in \mathbf{S}\}$$
$$\cap \{x' : (\overline{\mathbf{W}}_{i-1}(\mathbf{S})W_{ij})^T x' + r_{ij}(\mathbf{S}) \leq 0 \ \forall (i,j) \notin \mathbf{S}\} \cap \{x' : \overline{W}(\mathbf{S})^T x' + r(\mathbf{S}) \geq 0\},$$

thus matching the conditions of Lemma 5 when each $g_k$ function is affine. Furthermore, the set $J(x)$ is equal to the set of functions that are exactly zero at $x$, which consists of $\{(\overline{\mathbf{W}}_{i-1}(\mathbf{S})W_{ij})^T x + \overline{r}_{ij}(\mathbf{S}) : (i,j) \in T(\mathbf{x})\}$ together with $\overline{W}(\mathbf{S})^T x + \overline{r}(\mathbf{S})$. This observation combined with Lemma 5 gives the desired result. $\qquad\square$

Lemma 2 is a consequence of Proposition 2. For ease of exposition, we first reproduce the lemma and then present the proof.

**Lemma 6.** *The set $\mathcal{D}$ is positive invariant if and only if, for all $x \in \partial \mathcal{D}$, there exist $\mathbf{S} \in \mathbf{S}(x)$ and $u \in \mathcal{U}$ satisfying*

$$(\overline{\mathbf{W}}_{i-1}(\mathbf{S})W_{ij})^T (f(x) + g(x)u) \geq 0 \ \forall (i,j) \in \mathbf{T}(x) \cap \mathbf{S} \tag{23}$$

$$(\overline{\mathbf{W}}_{i-1}(\mathbf{S})W_{ij})^T (f(x) + g(x)u) \leq 0 \ \forall (i,j) \in \mathbf{T}(x) \setminus \mathbf{S} \tag{24}$$

$$(\overline{\mathbf{W}}_{i-1}(\mathbf{S})W_{ij})^T (f(x) + g(x)u) \geq 0 \tag{25}$$

*Proof.* By Theorem 3, the set $\mathcal{D}$ is positive invariant if and only if for every $x \in \partial \mathcal{D}$, there exists $u$ such that $(f(x) + g(x)u) \in \mathcal{T}_\mathcal{D}(x)$. By Proposition 2, this condition holds iff there exists $\mathbf{S} \in \mathbf{S}(x)$ such that

$$(f(x) + g(x)u) \in \left( \left( \bigcap_{(i,j) \in \mathbf{T}(x) \cap \mathbf{S}} \{z : (\overline{\mathbf{W}}_{i-1}(\mathbf{S})W_{ij})^T z \geq 0\} \right) \cap \right.$$
$$\left. \left( \bigcap_{(i,j) \in \mathbf{T}(x) \setminus \mathbf{S}} \{z : (\overline{\mathbf{W}}_{i-1}(\mathbf{S})W_{ij})^T z \leq 0\} \right) \cap \{z : \overline{W}(\mathbf{S})^T z \geq 0\} \right)$$

The above condition is equivalent to the conditions of the lemma, completing the proof. $\qquad\square$

### A.3 Proof of Proposition 1

First, suppose that condition (i) holds. Then for any $x \in \mathcal{D}$ with $\mathbf{S}(x) = \{\mathbf{S}_1, \ldots, \mathbf{S}_r\}$, there exists $l \in \{1, \ldots, r\}$ and $u \in \mathcal{U}$ such that $x \in \overline{\mathcal{X}}(\mathbf{S}_l)$ and (7)–(8) hold. For this choice of $u$, we have $(f(x) + g(x)u) \in \mathcal{T}_{\Psi_i}(x)$ by Proposition 2. Hence $\mathcal{D}$ is positive invariant under any control policy consistent with $b$ by Lemma 2.

Next, suppose that condition (ii) holds. Since $\mathcal{D}$ is contained in the union of the activation sets $\overline{\mathcal{X}}(\mathbf{S})$, this condition implies that $\mathcal{D} \subseteq \mathcal{C}$.

### A.4 Proof of Lemma 3

Suppose that condition 1 holds. Then for any $x \in \partial \mathcal{D}$ with $\mathbf{S}(x) = \{\mathbf{S}_1, \ldots, \mathbf{S}_r\}$, there exists $l \in \{1, \ldots, r\}$ such that $x \in \overline{\mathcal{X}}(\mathbf{S}_l)$ and $u \in \mathcal{U}$ satisfy (7) and (8). For this choice of $u$, we have $(f(x) + g(x)u) \in \mathcal{T}_\mathcal{D}(x)$ by Proposition 2. Hence $\mathcal{D}$ is positive invariant under any control policy consistent with $b$ by Theorem 3.

If Condition 2 holds, then there is no $x$ with $b(x) = 0$ and $x \in \text{int}(\overline{\mathcal{X}}(\mathbf{S}))$ such that $x \notin \mathcal{C}$. Hence, there are no counterexamples to condition (ii) of Proposition 1.

### A.5 Proof of Lemma 4

The approach is to prove that condition (ii) of Proposition 1 holds; condition (i) holds automatically if each $\mathbf{S}_1, \ldots, \mathbf{S}_r$ satisfies condition (ii) of Lemma 3. We have that conditions (a) and (b) are equivalent

to $b(x) = 0$ and (11). In order for $x$ to be a safety counterexample, for all $l = 1, \ldots, r$, at least one of Eqs. (7) and (8) must fail. Equivalently, for all $l = 1, \ldots, r$, there does not exist $u$ satisfying

$$
\begin{aligned}
-(\overline{\mathbf{W}}_{i-1}(\mathbf{S}_l)W_{ij})^T g(x)u &\leq (\overline{\mathbf{W}}_{i-1}(\mathbf{S}_l)W_{ij})^T f(x) \ \forall (i,j) \in T(\mathbf{S}_1, \ldots, \mathbf{S}_r) \cap \mathbf{S}_l \\
-(\overline{\mathbf{W}}_{i-1}(\mathbf{S}_l)W_{ij})^T g(x)u &\geq (\overline{\mathbf{W}}_{i-1}(\mathbf{S}_l)W_{ij})^T f(x) \ \forall (i,j) \in T(\mathbf{S}_1, \ldots, \mathbf{S}_r) \setminus \mathbf{S}_l \\
-\overline{W}_{ij}(\mathbf{S}_l)^T g(x)u &\leq \overline{W}(\mathbf{S}_l)^T f(x) \\
Au &\leq c
\end{aligned}
$$

By Farkas Lemma, non-existence of such a $u$ is equivalent to existence of $y_l$ satisfying $y_l \geq 0$ as well as (17) and (18).

## A.6 Proof of Theorem 2

Suppose that $x$ is a safety counterexample for the NCBF $b$ with $b(x) = 0$. If $x \in \text{int}\overline{\mathcal{X}}(\mathbf{S})$ for some $\mathbf{S}$, then we have that $\mathbf{S} \in \mathcal{S}$ and hence a contradiction with Lemma 3. If $x \in \overline{\mathcal{X}}(\mathbf{S}_1) \cap \cdots \overline{\mathcal{X}}(\mathbf{S}_r)$ for some $\mathbf{S}_1, \ldots, \mathbf{S}_r$, then there is a contradiction with Lemma 4.

## A.7 Details on the IBP Procedure

Interval bound propagation aims to compute an interval of possible output values by propagating a range of inputs layer-by-layer, and is integrated into our approach as follows. We first use partition the state space into cells and, for each cell, use LiRPA to derive upper and lower bounds on the value of b(x) when x takes values in that cell. When the interval of possible b(x) values in a cell contains zero, we conclude that that cell may intersect the boundary b(x) = 0. For each neuron, we use IBP to compute the pre-activation input interval for values of x within the cell. When the pre-activation input has a positive upper bound and negative lower bound, we identify the neuron as unstable, i.e., it may be either positive or negative for values of $x$ within the cell. Using this approach, we enumerate a collection of activation sets $\mathcal{S}$. We then identify the activation sets $\mathbf{S} \in \tilde{\mathcal{S}}$ such that $b(x) = 0$ for some $x \in \overline{\mathcal{X}}(\mathbf{S})$ by searching for an $x$ that satisfies the linear constraints in (16). This approach uses LiRPA and IBP to identify the activation regions that intersect the boundary $\{x : b(x) = 0\}$ without enumerating and checking all possible activation sets, which would have exponential runtime in the number of neurons in the network.

## A.8 Nonlinear Programming

The condition 2 of Lemma 3 suffices to solve the nonlinear program

$$
\begin{aligned}
\text{minimize} \quad & h(x) \\
\text{s.t.} \quad & \overline{W}_{ij}(\mathbf{S})^T x + \overline{r}_{ij}(\mathbf{S}) \geq 0 \ \forall (i,j) \in \mathbf{S} \\
& \overline{W}_{ij}(\mathbf{S})^T x + \overline{r}_{ij}(\mathbf{S}) \leq 0 \ \forall (i,j) \notin \mathbf{S} \\
& \overline{W}(\mathbf{S})^T x + \overline{r}(\mathbf{S}) = 0
\end{aligned}
\tag{26}
$$

and check whether the optimal value is nonnegative (unsafe) or negative (safe).

The verification problem of Lemma 4 can then be mapped to solving the nonlinear program

$$
\begin{aligned}
\min_{x, y_1, \ldots, y_r} \quad & \max_{l=1,\ldots,r} \left\{ y_l^T \Lambda_l(\mathbf{S}_1, \ldots, \mathbf{S}_r, x) \right\} \\
\text{s.t.} \quad & (\overline{\mathbf{W}}_{i-1}(\mathbf{S}_1)W_{ij})^T x + \overline{r}_{ij}(\mathbf{S}_1) < 0 \ \forall (i,j) \notin S_1 \cup \cdots \cup S_r \\
& (\overline{\mathbf{W}}_{i-1}(\mathbf{S}_1)W_{ij})^T x + \overline{r}_{ij}(\mathbf{S}_1) > 0 \ \forall (i,j) \in S_1 \cap \cdots \cap S_r \\
& (\overline{\mathbf{W}}_{i-1}(\mathbf{S}_1)W_{ij})^T x + \overline{r}_{ij}(\mathbf{S}_1) = 0 \ \forall (i,j) \in \mathbf{T}(\mathbf{S}_1, \ldots, \mathbf{S}_r) \\
& y_l^T \Theta_l(\mathbf{S}_1, \ldots, \mathbf{S}_r(x)) = 0 \ \forall l = 1, \ldots, r \\
& y_l \geq 0 \ \forall l = 1, \ldots, r
\end{aligned}
\tag{27}
$$

and checking whether the optimal value is nonnegative (safe) or negative (unsafe).

## A.9 Experiment Settings: Darboux

We show the settings of NCBF verification for Darboux system whose dynamic is defined as

$$
\begin{bmatrix} \dot{x}_1 \\ \dot{x}_2 \end{bmatrix} = \begin{bmatrix} x_2 + 2x_1 x_2 \\ -x_1 + 2x_1^2 - x_2^2 \end{bmatrix}.
\tag{28}
$$

We define state space, initial region, and unsafe region as $\mathcal{X} : \{\mathbf{x} \in \mathbb{R}^2 : x \in [-2,2] \times [-2,2]\}$, $\mathcal{X}_I : \{\mathbf{x} \in \mathbb{R}^2 : 0 \leq x_1 \leq 1, 1 \leq x_2 \leq 2\}$ and $\mathbf{x}_U : \{\mathbf{x} \in \mathbb{R}^2 : x_1 + x_2^2 \leq 0\}$ respectively.

## A.10 Experiment Settings: Obstacle Avoidance

We next evaluate that our proposed method on a controlled system [40]. The system state consists of 2-D position and aircraft yaw rate $x := [x_1, x_2, \psi]^T$. We let $u$ denote the control input to manipulate yaw rate and define the dynamics as

$$\begin{bmatrix} \dot{x}_1 \\ \dot{x}_2 \\ \dot{\psi} \end{bmatrix} = \begin{bmatrix} v\sin\psi \\ v\cos\psi \\ 0 \end{bmatrix} + \begin{bmatrix} 0 \\ 0 \\ u \end{bmatrix}. \tag{29}$$

We define the state space, initial region and unsafe region as $\mathcal{X}$, $\mathcal{X}_I$ and $\mathcal{X}_U$, respectively as

$$\mathcal{X} : \{\mathbf{x} \in \mathbb{R}^3 : x_1, x_2, \psi \in [-2,2] \times [-2,2] \times [-2,2]\}$$
$$\mathcal{X}_I : \{\mathbf{x} \in \mathbb{R}^3 : -0.1 \leq x_1 \leq 0.1, -2 \leq x_2 \leq -1.8, \ -\pi/6 < \psi < \pi/6\} \tag{30}$$
$$\mathcal{X}_U : \{\mathbf{x} \in \mathbb{R}^3 : x_1^2 + x_2^2 \leq 0.04\}$$

## A.11 Experiment Settings: Spacecraft Rendezvous

The state of the chaser is expressed relative to the target using linearized Clohessy–Wiltshire–Hill equations, with state $x = [p_x, p_y, p_z, v_x, v_y, v_z]^T$, control input $u = [u_x, u_y, u_z]^T$ and dynamics defined as follows.

$$\begin{bmatrix} \dot{p}_x \\ \dot{p}_y \\ \dot{p}_z \\ \dot{v}_x \\ \dot{v}_y \\ \dot{v}_z \end{bmatrix} = \begin{bmatrix} 1 & 0 & 0 & 0 & 0 & 0 \\ 0 & 1 & 0 & 0 & 0 & 0 \\ 0 & 0 & 1 & 0 & 0 & 0 \\ 3n^2 & 0 & 0 & 0 & 2n & 0 \\ 0 & 0 & 0 & -2n & 0 & 0 \\ 0 & 0 & -n^2 & 0 & 0 & 0 \end{bmatrix} \begin{bmatrix} p_x \\ p_y \\ p_z \\ v_x \\ v_y \\ v_z \end{bmatrix} + \begin{bmatrix} 0 & 0 & 0 \\ 0 & 0 & 0 \\ 0 & 0 & 0 \\ 1 & 0 & 0 \\ 0 & 1 & 0 \\ 0 & 0 & 1 \end{bmatrix} \begin{bmatrix} u_x \\ u_y \\ u_z \end{bmatrix}. \tag{31}$$

We define the state space and unsafe region as $\mathcal{X}$ and $\mathcal{X}_U$, respectively as

$$\mathcal{X} : \{\mathbf{x} \in \mathbb{R}^3 : p, v, \in [-1.5, 1.5] \times [-1.5, 1.5]\}$$
$$\mathcal{X}_U : \left\{ 0.25 \leq r \leq 1.5, \text{ where } r = \sqrt{p_x^2 + p_y^2 + p_z^2} \right\} \tag{32}$$

We obtain the trained NCBF with neural CLBF training in [13] with a nominal model predictive controller.

## A.12 Experiment Settings: hi-ord$_8$

The dynamic model of the system is captured by an ODE as follows.

$$x^{(8)} + 20x^{(7)} + 170x^{(6)} + 800x^{(5)} + 2273x^{(4)} + 3980x^{(3)} + 4180x^{(2)} + 2400x^{(1)} + 576 = 0 \tag{33}$$

where we denote the $i$-th derivative of variable $x$ by $x^{(i)}$. We define the state space and unsafe region as $\mathcal{X}$ and $\mathcal{X}_U$, respectively as

$$\mathcal{X} : \{x_1^2 + \ldots + x_8^2 \leq 4\}$$
$$\mathcal{X}_U : \left\{ (x_1 + 2)^2 + \ldots + (x_8 + 2)^2 \leq 0.16 \right\} \tag{34}$$

We obtain the trained NCBFs with training method proposed in [21].

## A.13 Training Details

We trained the NCBFs for Darboux and obstacle avoidance via the approach proposed in [24]. Models are trained with their open source code [1] with default settings. Detailed parameters for both cases listed in Table 3a.

---

[1] `https://github.com/zhaohj2017/HSCC20-Repeatability`

We then trained NCBFs for Spacecraft Rendezvous by following approach proposed in [13] with the empirical loss defined in Eq. (5) in [12]. Models are trained with their open source code [2] with default settings. The hyper-parameters are listed in Table 3b.

Table 3: Hyper-parameters for training NCBFs to be verified

(a) Hyper-parameters for Darboux and OA

| Hyper-Parameters | Value |
| --- | --- |
| LEARNING_RATE | 0.01 |
| LOSS_OPT_FLAG | 1e-16 |
| TOL_MAX_GRAD | 6 |
| EPOCHS | 5 |
| TOL_INIT | 0.0 |
| TOL_SAFE | 0.0 |
| TOL_BOUNDARY | 0.05 |
| TOL_LIE | 0.0 |
| TOL_NORM_LIE | 0.0 |
| WEIGHT_LIE | 1 |
| WEIGHT_NORM_LIE | 0 |
| DECAY_LIE | 1 |
| DECAY_INIT | 1 |
| DECAY_UNSAFE | 1 |

(b) Hyper-parameters for Spacecraft Randezvous

| Hyper-Parameters | Value |
| --- | --- |
| LEARNING_RATE | 0.01 |
| BATCH_SIZE | 512 |
| CONTROLLER_PERIOD | 0.01 |
| SIMULATION_DT | 0.01 |
| CBF_HIDDEN_LAYERS | 1 |
| CBF_HIDDEN_SIZE | 16 |
| CBF_LAMBDA | 0.1 |
| CBF_RELAXATION_PEN | 1e4 |
| SCALE_PARAMETER | 10.0 |
| PRIMAL_LEARNING_RATE | 1e-3 |
| LEARN_SHAPE_EPOCHS | 100 |

## A.14 Experiment Details and Results

We use translators and verifiers proposed in FOSSIL[3] for SMT-based verification with solver dReal and Z3 as baselines. Our proposed enumerating algorithm utilize auto-LiRPA [4] with default settings and linear program with HiGHS solvers provided by SciPy [5]. Detailed settings can be found in our attached code.

We further visualize the trend of the number of activation sets and run-time with respect to the total number of neurons with ReLU activation function in Fig. 3. We can find that the logarithm of the activation sets size grows with the size of the neural network. The dimensionality of the state is the dominant factor in determining the run-time. The logarithm of the run-time is determined by both the state dimension and the number of ReLU hidden layers. The potential result can be cause by loose activation set estimation. Methods deriving tighter bounds than IBP may mitigate the influence of the ReLU hidden layers.

## A.15 Comparison of NCBF and SOS-based Synthesis

We compare NCBF with traditional SOS synthesized polynomial CBF for the obstacle avoidance case study in two aspects, namely, training time $T_t$ and volume $V$ of the guaranteed safe region. In order to synthesize the polynomial CBF, we adopt the procedure introduced in [45]. This procedure first constructs a nominal controller $\mu(x)$, and then uses SOS programming to construct a barrier certificate for the system $\dot{x}(t) = f(x) + g(x)\mu(x)$. We choose $\mu(x) = -x_3$ as the nominal controller and synthesize CBFs of degree 2, 4, 6, 8, and 10 using the Matlab SOSTOOLS toolbox. We compared the result with an NCBF with one hidden layer of 32 neurons trained using the method proposed in [24] with the same nominal controller. The experiment results are shown below. The time of SOS CBF synthesis grows with the degree of the barrier function. Degree 10 CBF takes twice the time compared to NCBF. On the other hand, NCBF outperforms all SOS synthesized CBFs by having the largest safe region volume.

---

[2] https://github.com/MIT-REALM/neural_clbf

[3] https://github.com/oxford-oxcav/fossil

[4] https://github.com/Verified-Intelligence/auto_LiRPA

[5] https://docs.scipy.org/doc/scipy-1.10.1/reference/optimize.linprog-highs.html

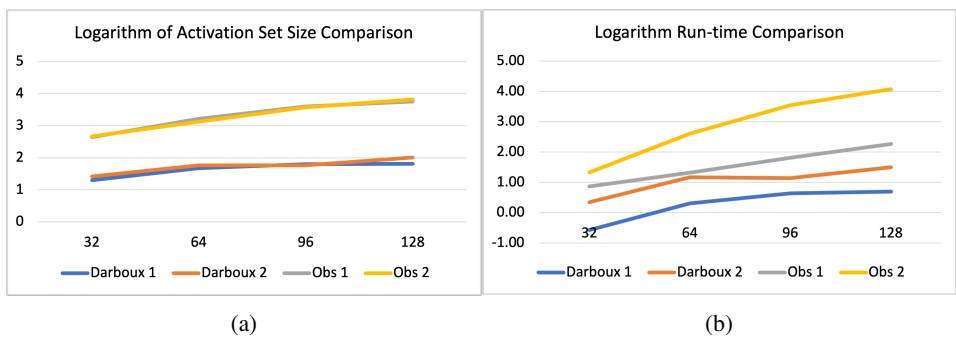

(a)                                                    (b)

Figure 3: Comparison of the number of activation sets and run-time with respect to number of neurons in total. (a) shows logarithm of activation set size. (b) shows logarithm of run-time. We denote NCBFs for Darboux with 1 and 2 hidden layers as Darboux 1 and Darboux 2, respectively. We denote NCBFs for obstacle avoidance with 1 and 2 hidden layers as obs 1 and obs 2, respectively.

Table 4: Comparison of the training time $T_t$ and safe region volume $V$ of a NCBF and SOS synthesized CBFs for Obstacle Avoidance

| Types | $T_t\ (s)$ | $V\ (m^2 \times deg)$ |
|---|---|---|
| NCBF 3-32-$\sigma$-1 | 262.89s | 37.76 |
| SOS Degree 2 | 7.36s | 16.14 |
| SOS Degree 4 | 6.65s | 13.44 |
| SOS Degree 6 | 19.88s | 31.36 |
| SOS Degree 8 | 125.10s | 25.93 |
| SOS Degree 10 | 551.31s | 19.99 |

## A.16  Comparison between Activation Functions

We considered three case studies, namely, Darboux, obstacle avoidance, and spacecraft rendezvous. For each case study, we trained and verified three NCBFs with the same architecture (2 hidden layers of 32 neurons each) but different activation functions, namely, ReLU, sigmoid, and tanh. We found

Table 5: Comparison of training time $T_t$, safety region volume $V$ and verification time $T_v$ of ReLU, Sigmoid and Tanh NCBF for Darboux, Obstacle Avoidance and Spacecraft Rendezvous. ReLU NNs are verified by proposed method while others are verified by dReal and Z3. We write UTD when the method cannot be not directly used for verification

| Case | Darboux | | | Obstacle Avoidance | | | Spacecraft Rendezvous | |
|---|---|---|---|---|---|---|---|---|
| | ReLU | Sigmoid | Tanh | ReLU | Sigmoid | Tanh | ReLU | Tanh |
| $T_t$ | 28.53s | 51.14s | 69.49s | 71.44s | 78.66s | 76.49s | 879.388s | 953.469s |
| $V$ | 2.27 | 1.62 | 2.8 | 4.99 | 2.17 | 3.50 | 0.20 | 0.22 |
| $T_v$ | 14.64 | >3hrs | >3hrs | 273.37s | >3hrs | >3hrs | 13906.19s | UTD |

that, for the Darboux and obstacle avoidance case studies, the ReLU NCBF completed training faster than both sigmoid and tanh NCBFs. The volume of the safe region was comparable for all three activation functions, with the tanh outperforming the ReLU NCBF in Darboux and the ReLU NCBF providing the largest volume for obstacle avoidance. The most significant difference between the three activation functions was at the verification stage. Our proposed method for verifying ReLU NCBFs terminated within 15 and 274 seconds in the Darboux and obstacle avoidance, respectively, while SMT-based methods did not terminate within three hours for both test cases. In the spacecraft rendezvous example, the ReLU NCBF completed training before the tanh NCBF. Moreover, while our approach verified the correctness of the ReLU NCBF within 4 hours, the tanh NCBF exhibited a safety violation.

## A.17   Example Details

Consider the setting of the example in Section 3.2. Let $b_c$ denote the NCBF defined in the example, which fails our defined safety conditions. For comparison, we trained an NCBF $b_\theta$ and verified it using our proposed approach. We then constructed a nominal controller $\mu_{nom}$ as a Linear Quadratic Regulator (LQR) controller that drives the system from initial point $(0, 0.1)$ to the origin. We compared the trajectories arising from the optimization-based controller defined by Eq. (10) using the $b_\theta$ and $b_c$. For the unsafe NCBF $b_c$, the optimization-based controller is unable to satisfy the safety constraints at the boundary point $(0, 1)$, resulting in a safety violation. On the other hand, while the NCBF $b_\theta$ contained multiple non-differentiable points, it is possible to choose $u$ to ensure safety at these points. For example, the point $(-0.19, 2.91)$ is a non-differentiable point on the boundary $b_\theta = 0$. There are four activation sets intersecting at this point, with corresponding values of $\frac{\partial b_c}{\partial x} g(x)$ given by $\{-0.0455, -0.053, -0.025, -0.033\}$. Since any control input $u$ with negative sign and sufficiently large magnitude will satisfy $\frac{\partial b_c}{\partial x}(f(x) + g(x)u) \geq 0$ for all of these values, this non-differentiable point does not compromise safety of the system, and the trajectory of the system constrained by $b_\theta$ remains in the safe region for all time.

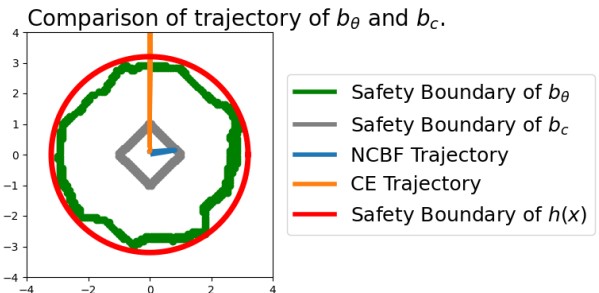

Figure 4: Comparison of optimization-based controller using trained NCBF $b_\theta$ and unsafe NCBF $b_c$.

