# OpenReview forum: "Exact Verification of ReLU Neural Control Barrier Functions"
_NeurIPS.cc/2023/Conference — NeurIPS 2023 poster_

### Official Review · Reviewer_hbj4 · 2023-07-02

**Soundness:** 3 good
**Presentation:** 2 fair
**Contribution:** 2 fair
**Rating:** 6
**Confidence:** 3

**Summary:**

This paper studies the exact conditions under which a learned CBF $b$ with ReLU activations yields the positive invariance property. Even though the set of inputs for which the CBF is non-differentiable has measure zero, the paper shows by example that safety can be still violated. This is because the slope of $b(x)$ can be discontinuous at the boundary $b(x) = 0$. To address this, the paper derives a theoretical result (Proposition 1) that provides a necessary and sufficient condition for a valid ReLU CBF. Based on this central result, the paper presents a verification algorithm that consists of state space discretization and search, activation set enumeration, and solving nonlinear programs. The experiments show faster safety verification for low-dimensional systems than SMT-based approaches.

**Strengths:**

* ReLU is a popular activation function that is often used for constructing feedforward neural networks. The paper addresses the important theoretical problem of verifying learned CBFs with ReLU activations.
* The paper provides a novel, central theoretical result (Proposition 1) that gives necessary and sufficient conditions for a given ReLU neural CBF to be valid.
* As shown in Section 5, the proposed verification algorithm yields more efficient run-time than SMT-based approaches.


**Weaknesses:**

* From a practical point of view, an engineer or a researcher can always choose a differentiable activation function to design a neural CBF, such as tanh or softplus. Then, the CBF remains differentiable. This leads to a motivational question of why we should use ReLU activations for modeling CBFs in the first place. Specifically, the proposed approach seems to possess several disadvantages compared to learning differentiable CBFs and verifying them. 1) The verification algorithm involves state-space discretization and activation set enumeration, both of which have inherent poor scalability with respect to the state-space dimensionality and the size of the neural network, respectively. In particular, the enumeration seems to suffer a lot as the network becomes complex, as observed in Table 2. 2) The online optimization problem for control (i.e. equation 10) requires solving possibly multiple quadratic programs, adding complexity to the standard CBF-QP.

* The presentation can be improved so that mathematical descriptions are easier to follow. Especially, it is recommended that the paper 1) clearly define dimensionality of every vector and matrix-valued variables introduced in the paper, such as $W_i$, $W_{ij}$. 2) Define clearly any non-trivial variables in the main statement of mathematical results. For instance, the definition of $\bar{W}_0$ and $\bar{r}_0$ should be provided in the main statement of Lemma 1, not in the proofs in the appendix.


**Questions:**

* What are the theoretical and practical advantages of using ReLU activations over differentiable activations (as done in [1], for example) for modeling neural CBFs?

* Shouldn't the last formula in the proof of Lemma 1 use $W_{Lj}$, not $W_{ij}$?

[1] Dawson, Charles, Zengyi Qin, Sicun Gao, and Chuchu Fan. "Safe nonlinear control using robust neural lyapunov-barrier functions." In Conference on Robot Learning, pp. 1724-1735. PMLR, 2022.

**Limitations:**

* The most concerning limitation is the scalability of the verification method. Even if the state space has a low-dimensionality, the usage of more layers or neurons can lead to combinatorial explosion of the runtime complexity.

---

> ### Author Rebuttal · Authors · 2023-08-09
>
> We conducted an experimental study to compare NCBFs with different activation functions, including ReLU, sigmoid, and tanh. The results are shown in the pdf attachment to the general rebuttal, and are discussed in detail in the rebuttal to Reviewer wvvT above. In particular, we find that our proposed algorithm for verifying ReLU-activated NCBFs is more computationally tractable compared to SOTA algorithms for verifying NCBFs with differentiable activation functions.
>
> The reviewer is correct that the online optimization-based control requires solving multiple quadratic programs in the worst-case. However, there will only be multiple quadratic programs at points where the NCBF is nondifferentiable, or equivalently, where the pre-activation input to one of the neurons in the NCBF is exactly zero. In order to evaluate the complexity of the CBF-QP, we compared the computation time of (i) Eq. (10) with a NCBF consisting of one hidden layer of 32 neurons and (ii) a CBF-QP using a degree-two polynomial CBF. Averaging over 300 iterations, we found that the NCBF-based controller had a runtime of 0.0015s while the polynomial CBF-based controller had a mean runtime of 0.0012s.
>
> $W_{ij}$ has dimensionality $n \times 1$ for $i=1$ and $M_{i-1} \times 1$ for $i > 1$. $W_{i}$ is an $n \times M_{i}$ matrix. We will clearly define matrix and vector dimensions throughout the paper, as well as provide definitions of $\overline{W}\_{0}$ and $\overline{r}\_{0}$ in the main statement of the lemma. The reviewer is correct regarding the last formula in the proof of Lemma 1. It should be $W_{Lj}$.

---

> > ### Comment · Reviewer_hbj4 · 2023-08-13
> > **Acknowledgement**
> >
> > Thank you for your convincing response. The additional experiments are interesting in that the ReLU activation results in faster learning and verification than other commonly-used functions, such as tanh. This indeed motivates the use for ReLU activations for NCBFs.
> > Based on Table 2 and 3 in the new PDF, it seems that the proposed algorithm has better scalability than existing methods, which is an encouraging result.
> > Thank you also for clarifying the condition under which the worst-case solve time for QP occurs.

---

> > > ### Author Response · Authors · 2023-08-13
> > >
> > > We thank the reviewer for their time and effort providing feedback on the manuscript and rebuttal.

---

### Official Review · Reviewer_yuYJ · 2023-07-03

**Soundness:** 3 good
**Presentation:** 2 fair
**Contribution:** 3 good
**Rating:** 6
**Confidence:** 4

**Summary:**

Neural control barrier functions offer wider expressive power but do not satisfy the continuously differentiable assumption with ReLU activation functions. In this paper, the authors propose a method to verify that a neural ReLU CBF is a valid CBF and is hence capable of rendering a set forward-invariant. This method derives its foundations by extending the conditions of Nagumo’s theorem. These observations provide a set of necessary and sufficient conditions for verification. Algorithmically, these conditions are verified by a combination of linear relaxation to do a coarse analysis followed by interval bound propagation to fine-tune the regions of violation. Then, two non-linear programs are used to check for the CBF conditions. The final verification engine runs faster than existing SMT methods and works correctly on three benchmarks: Darboux, Obstacle avoidance and Spacecraft.

**Strengths:**

The theoretical conditions for verification and the proposed algorithm are novel, principled, creative and non-trivial. Previous works typically gloss over the points of non-differentiability in energy functions such as CLF/CBF. Hence, I believe this is an important contribution which can be extended further in future work.

**Weaknesses:**

1) The presentation needs to be improved and there are several minor mistakes which put off the reader.
Line 243 - missing reference, Line 181 - mistake in the definition of complete collection, Line 439 -  What definition of distance is used in the definition of tangent cone?, Multiple typos in Lemma 5 in appendix

2) If the set of non-differentiable points has measure zero, it will not actually affect the practical safety filtering application as the system can shoot through those points.

3) For the experiments section, it would be interesting to also plot the points of non-differentiability in figure 2. The current result comparing to improperly trained CBF is rather expected. Showing the enhancement of safety verification/safety filtering by virtue of considering non-differentiable points will enhance the contribution.


**Questions:**

1) In proposition 1, equation (11), is the set difference explicitly needed given that the collection of sets $\{ S_1, \dots, S_r \}$ is defined as complete.

2) What is the meaning of equation (15) in line 186? Why can that not just be written as $\mathcal{D} \subset \mathcal{C}$ ?

3) For the procedure discussed in figure 1, I did not find a lot of background material in the appendix on exactly how IBP is used to enumerate activated sets and which linear program is used for pruning. I recommend a discussion on this in greater detail in the Appendix similar to Section 7.7 for the non-linear program. This is the phase that is seen to be time-consuming from Table 2 and needs to be justified.


**Limitations:**

The societal impact of this work is likely positive.
Limitations on the scalability of the approach to higher dimensional systems and deeper networks is discussed.

---

> ### Author Rebuttal · Authors · 2023-08-09
>
> We thank the reviewer for identifying some minor mistakes in the manuscript. The missing reference refers to Eq. (27) from the supplementary material and will be corrected. The definition of complete collection will be modified to $\cdots \cap \overline{\mathcal{X}}(\mathbf{S}^{\prime})$. Distance is defined by $\mbox{dist}(x,\mathcal{C}) = \min{\\{||x-z|| : z \in \mathcal{C}\\}}$, where $||\cdot||$ is a $p$-norm for some $p \in [1,\ldots,\infty]$. In Lemma 5, the definition of $J(x)$ should be $J(x) = \\{k: q_k(x) = 0\\}$, while the following equation should be $\mathcal{T}\_{\mathcal{A}}(x) = \\{z : z^{T}\nabla q_{k}(x) \leq 0 \ \forall k \in J(x)\\}.$
>
> The reviewer is correct that the set of non-differentiable points will have measure zero. As shown in the example on Page 4, although this set has measure zero, safety of the CBF cannot be guaranteed if the non-differentiable points fail the conditions (7)--(9). When applying the safety filter at runtime, the optimization problem (10) reduces to a standard CBF-based quadratic program, except at the measure-zero set of non-differentiable points. If non-differentiable points occur in the interior of the region $\mathcal{D} = \\{x : b(x) \geq 0\\}$, we conjecture that it may be possible to shoot through them as suggested by the reviewer. Safety violations may occur if the controller attempts to shoot through non-differentiable points at the boundary, as shown in the following example.
>
> Consider the setting of the example on Page 4 of the manuscript. Let $b_{c}$ denote the NCBF defined in the example, which fails our defined safety conditions. For comparison, we trained an NCBF $b_{\theta}$ and verified it using our proposed approach. We then constructed a nominal controller $\mu_{nom}$ as a Linear Quadratic Regulator (LQR) controller that drives the system from initial point $(0,0.1)$ to the origin. We compared the trajectories arising from the optimization-based controller defined by Eq. (10) using the $b_{\theta}$ and $b_{c}$. For the unsafe NCBF $b_{c}$, the optimization-based controller is unable to satisfy the safety constraints at the boundary point $(0,1)$, resulting in a safety violation as described in the manuscript. On the other hand, while the NCBF $b_{\theta}$ contained multiple non-differentiable points, it is possible to choose $u$ to ensure safety at these points. For example, the point $(-0.19, 2.91)$ is a non-differentiable point on the boundary $b_{\theta} = 0$. There are four activation sets intersecting at this point, with corresponding values of $\frac{\partial b_{c}}{\partial x}g(x)$ given by $\\{-0.0455, -0.053, -0.025, -0.033\\}$. Since any control input $u$ with negative sign and sufficiently large magnitude will satisfy $\frac{\partial b_{c}}{\partial x}(f(x)+g(x)u) \geq 0$ for all of these values, this non-differentiable point does not compromise safety of the system, and the trajectory of the system constrained by $b_{\theta}$ remains in the safe region for all time.
>
> The reviewer is correct that the set difference is not needed in Eq. (11). It will be removed. Eq. (15) is presented to specify the conditions that must hold for all activation set $\mathbf{S}$. After taking the union over $\mathbf{S}$, the statement is equivalent to $\mathcal{D} \subset \mathcal{C}$ as pointed out by the reviewer.
>
> Interval bound propagation aims to compute an interval of possible output values by propagating a range of inputs layer-by-layer, and is integrated into our approach as follows. We first use partition the state space into cells and, for each cell, use CROWN to derive upper and lower bounds on the value of b(x) when x takes values in that cell. When the interval of possible b(x) values in a cell contains zero, we conclude that that cell may intersect the boundary b(x) = 0. For each neuron, we use IBP to compute the pre-activation input interval for values of x within the cell. When the pre-activation input has a positive upper bound and negative lower bound, we identify the neuron as unstable, i.e., it may be either positive or negative for values of $x$ within the cell. Using this approach, we enumerate a collection of activation sets $\mathcal{S}$. We then identify the activation sets $\mathbf{S} \in \tilde{\mathcal{S}}$ such that $b(x) = 0$ for some $x \in \overline{\mathcal{X}}(\mathbf{S})$ by searching for an $x$ that satisfies the linear constraints in (16). This approach uses CROWN and IBP to identify the activation regions that intersect the boundary $\\{x: b(x) = 0\\}$ without enumerating and checking all possible activation sets, which would have exponential runtime in the number of neurons in the network. We will add a section to the appendix that elaborates on IBP and its use in our verification algorithm.

---

> > ### Comment · Reviewer_yuYJ · 2023-08-13
> > **Read author response**
> >
> > I have read the author's response to my comments. The response is convincing. Further, additional supplementary experiments have been provided to show that ReLU activations are more beneficial and can potentially give a larger RoA. These results are encouraging. The only concern that remains for me are the tractable relaxations for verification that have recently come to my knowledge through the other reviewers.

---

> > > ### Author Response · Authors · 2023-08-13
> > >
> > > We thank the reviewer for the detailed feedback and for considering our rebuttal. We emphasize that the relaxations considered in the related works highlighted by Reviewer U7BK were developed for discrete-time systems and are not applicable to the continuous-time setting of our manuscript. Verification of continuous-time NCBFs raises new technical issues, for example, the non-differentiability of the NCBF with ReLU activation, that we address in the paper.

---

### Official Review · Reviewer_U7BK · 2023-07-06

**Soundness:** 3 good
**Presentation:** 3 good
**Contribution:** 2 fair
**Rating:** 6
**Confidence:** 3

**Summary:**

The authors consider the problem of synthesising control barrier functions parametrised as ReLU neural networks for non-linear deterministic dynamical systems. The authors first extend standard approaches for synthesising control barrier functions to the case where the barrier function is non-differentiable in a zero-measure set. Then, they encode the synthesis of a neural control barrier function as a nonlinear optimisation problem and illustrate the effectiveness of the approach on three benchmarks

**Strengths:**

-	The theory is sound and the extension of standard results for synthesizing control barrier functions to non-differentiable barriers is surely of interest
-	Paper is overall well written and the problem considered of interest


**Weaknesses:**

-	The main weakness is the scalability of the approach wrt the complexity of the neural barrier function. While I acknowledge that the algorithm is superior in terms of scalability than SMT based approaches, still experimental results are limited to neural networks of 1 hidden layer and 20 neurons at most
-	Some of the statements in the related works are not precise. In fact, the authors claim that it is not possible to use convex programming to synthesize/verify neural barrier functions. This is not accurate, in fact recent approaches rely on piecewise linear uncertain relaxations of neural networks to encode the problem of verifying neural barrier functions using linear programming [Mathiesen, Frederik Baymler, et al. "Safety certification for stochastic systems via neural barrier functions." IEEE Control Systems Letters 7 (2022): 973-978.] or SDP [Mazouz, Rayan, et al. "Safety guarantees for neural network dynamic systems via stochastic barrier functions." Advances in Neural Information Processing Systems 35 (2022): 9672-9686.]. Of course, the resulting approach will be more conservative compared to the one proposed by the authors, but more scalable. Also, please expand the discussion of why approaches employed to synthesize neural Lyapunov functions, e.g. [Abate, Alessandro, et al. "Formal synthesis of Lyapunov neural networks." IEEE Control Systems Letters 5.3 (2020): 773-778], cannot be employed in the setting of this paper
-	To demonstrate the importance of using neural control barrier functions, I believe that in the experiments there should be at least one experiment where the authors compare with the standard approaches commonly employed to synthesise control barrier functions, e.g. parametrising them as a SoS polynomial. This should serve to empirically demonstrate the advantages in being able to make use of the flexibility of neural networks in the context of barrier functions.


**Questions:**

Please, see Weaknesses Section and in addition consider the following points:

- Equation (7) and (8) quantifies over pairs $(i, j)$ of the set $\mathbf{T}(x) \cap \mathbf{S}$. However, $\mathbf{T}(x)$ is the set of unstable neurons produced by input $x$, hence it is unclear what pairs the quantification is referring to. Is it any two unstable neurons in the set $\mathbf{T}(x) \cap \mathbf{S}$?

- Set $\bar{\mathcal{X}}(\mathbf{S})$ is defined both inside Lemma 1 and right after the Lemma. Please, be consistent


**Limitations:**

Please, see Weaknesses Section

---

> ### Author Rebuttal · Authors · 2023-08-09
>
> In order to evaluate our approach on more complex neural networks, we conducted additional experiments as shown in Tables 2 and 3 of the pdf attachment to the general rebuttal. We verified an NCBF for the two-dimensional Darboux example with two hidden layers of 512 neurons each. The verification terminated successfully in 620 seconds. We verified an NCBF with two hidden layers of 64 neurons each for the three-dimensional obstacle avoidance example in 3 hours, 15 minutes. Finally, we verified an NCBF with three hidden layers, each containing 32 neurons, for a six-dimensional spacecraft rendezvous problem within four hours. For each of these networks, state of art verification algorithms dReal and Z3 did not terminate within three hours.
>
> The reviewer is correct that prior works have developed computationally tractable relaxations of the problem of verifying neural Lyapunov and barrier functions, whereas the aim of our paper is to develop exact conditions and algorithms. Moreover, we note the following key differences between our problem setting and the problems studied in the works identified by the reviewer. First, the prior works consider a discrete-time setting. Developing analogous conditions in continuous time requires addressing the non-differentiability of the gradient of the neural network activation functions, which is a contribution of the present paper. Second, the prior works assume that a control law has been given, while our proposed approach can be applied as a safety filter to an arbitrary given controller via the optimization problem (10).
>
> To reduce confusion, we will change the last sentence of the first paragraph of the related work to “However, SOS-based approaches for polynomial CBFs cannot be applied directly to NCBF verification, since activation functions used in neural networks are not polynomial and may be non-differentiable.” We will also add a sentence to the second paragraph of the related work that reads “Piecewise linear approximations of ReLU neural networks have been used to develop tractable safety verification algorithms using linear and SOS programming. This approach leads to sound and incomplete verification algorithms, whereas the present paper proposes exact verification algorithms. Moreover, these existing works apply to discrete-time systems with a priori given controllers.” We will add citations to the papers suggested by the reviewer.
>
> We compare NCBF with traditional SOS synthesized polynomial CBF for the obstacle avoidance case study in two aspects, namely, training time $T_t$ and volume $V$ of the guaranteed safe region. In order to synthesize the polynomial CBF, we adopt the procedure of “SOSTOOLS and its Control Applications” (Prajna et al). This procedure first constructs a nominal controller $\mu(x),$ and then uses SOS programming to construct a barrier certificate for the system $\dot{x}(t) = f(x) + g(x)\mu(x)$. We choose $\mu(x)=-x_{3}$ as the nominal controller and synthesize CBFs of degree 2, 4, 6, 8, and 10 using the Matlab SOSTOOLS toolbox. We compared the result with an NCBF with one hidden layer of 32 neurons trained using the method proposed in [34] with the same nominal controller.
> The experiment results are shown below. The time of SOS CBF synthesis grows with the degree of the barrier function. Degree 10 CBF takes twice the time compared to NCBF. On the other hand, NCBF outperforms all SOS synthesized CBFs by having the largest safe region volume.
>
>
> | CBF Type                | $T_t \ (s)$ | $V \ (m^2\times deg)$ |
> |----------------------|-------------|-----------------------|
> | NCBF 3-32-$\sigma$-1 | 262.89s     | 37.76                 |
> | SOS Degree 2         | 7.36s       | 16.14                 |
> | SOS Degree 4         | 6.65s       | 13.44                 |
> | SOS Degree 6         | 19.88s      | 31.36                 |
> | SOS Degree 8         | 125.10s     | 25.93                 |
> | SOS Degree 10        | 551.31s     | 19.99                 |
>
> In Eqs. (7) and (8), $(i,j)$ refers to the $j$-th neuron at the $i$-th layer. We will remove redundant definitions from the final manuscript.

---

> > ### Comment · Reviewer_U7BK · 2023-08-13
> > **Response to Rebuttal**
> >
> > I thank the authors for the rebuttal and the additional experiments, which clarified some of my doubt and the contributions. Consequently, I increase my score to weak accept.
> >
> > - We will also add a sentence to the second paragraph of the related work that reads “Piecewise linear approximations of ReLU neural networks have been used to develop tractable safety verification algorithms using linear and SOS programming. This approach leads to sound and incomplete verification algorithms, whereas the present paper proposes exact verification algorithms. Moreover, these existing works apply to discrete-time systems with a priori given controllers.
> >
> > I would put the emphasis especially on the fact that you focus on exact verification algorithms, while these relaxations will only produce sound and incomplete results and they have only be applied for discrete-time systems. In fact, controller synthesis for these alternative approaches has been considered, see e.g., Section 5 in [Mazouz, Rayan, et al. "Safety guarantees for neural network dynamic systems via stochastic barrier functions." Advances in Neural Information Processing Systems 35 (2022): 9672-9686.]

---

> > > ### Author Response · Authors · 2023-08-14
> > >
> > > We thank the reviewer for the feedback. We will follow the reviewer's recommendation when revising the related work and presenting the contributions of the paper.

---

### Official Review · Reviewer_wvvT · 2023-07-26

**Soundness:** 3 good
**Presentation:** 3 good
**Contribution:** 3 good
**Rating:** 6
**Confidence:** 2

**Summary:**

In this paper, the authors present a new strategy for verifying neural control barrier functions (NCBFs) to ensure safe control of nonlinear systems. Specifically, the authors address the challenge posed by NCBFs when ReLU activation functions are employed. This renders existing verification strategies inapplicable due to the non-differentiability of resulting NCBFs. The authors tackle this issue by introducing conditions for a ReLU NCBF to satisfy \emph{positive invariance}, which forms the basis for constructing safe control policies. The proposed verification algorithm enumerates \emph{activation sets} and individually verifies their positive invariance defined based on the positivity of the activation outputs. The paper demonstrates the effectiveness of the proposed approach by comparing it with Satisfiability Modulo Theory (SMT) based methods across three control problems.

**Strengths:**

- The paper contributes to the field by introducing a new verification algorithm and control strategies that apply to NCBFs using ReLU activation functions. Existing verification approaches are not directly applicable, and the authors address this limitation by introducing a new characterization of active sets and a corresponding verification algorithm. This approach has the potential to broaden the application domain of neural control barrier functions.
- The paper appears to be technically sound.

**Weaknesses:**

- While the comparison with SMT-based methods is informative, it would be beneficial to include a more comprehensive performance evaluation of ReLU NCBFs against classical NCBFs with differentiable activations, such as sigmoid. Assessing factors like NCBF training time, verification time, and the expressive capabilities of the resulting NCBFs in real-world control problems would add depth to the analysis. If ReLU NCBFs consistently outperform sigmoid NCBFs, the proposed verification strategy's utility would be significantly enhanced.
- The authors could provide a thorough analysis of the time complexity of the proposed verification strategy. As the algorithm discretizes the state space into hyper-cubes and calculates corresponding bounds of $b(x)$ individually, it is important to understand how the complexity scales in high-dimensional problems. Also, the authors should provide experiments on more complex, potentially higher-dimensional control problems.
- (Minor) the manuscript contains several typos that require careful proofreading.

**Questions:**

N/A

**Limitations:**

The authors should consider the possibility of clearly outlining the limitations of their study and potential negative societal impacts.

After the rebuttal phase (the same as the post-rebuttal comments):
I thank the authors for their response and conducting the additional experiments. The majority of the concerns raised in my initial evaluation have been effectively attended to. While I acknowledge the efforts undertaken by the authors to conduct an additional round of experiments, I still have reservations about how well the proposed algorithm would work for problems with high dimensions. The additional experiments provided are still confined to low-dimensional scenarios. I recommend that the authors explore this aspect in their future research endeavors.

---

> ### Author Rebuttal · Authors · 2023-08-09
>
> In order to compare with NCBFs that have differentiable activation functions, we conducted the following additional simulation study. We considered three test systems, namely, the Darboux, obstacle avoidance, and spacecraft rendezvous test cases. For the Darboux and obstacle avoidance  systems, we trained three NCBFs with the same architecture, i.e., 2 hidden layers and 32 neurons in each hidden layer, but different activation functions. The chosen activation functions were ReLU, Sigmoid, and tanh. We compared the performance based on three metrics: (i) training time, defined as the time required for the running loss to converge to 0, (ii) volume of the safe region $\{x: b(x) \geq 0\}$, and (iii) time required to verify the trained NCBF. We verified the ReLU-activated NCBF using our proposed method and verified the Sigmoid and tanh-activated NCBFs using dReal and Z3.
>
> We trained NCBFs for spacecraft rendezvous using the NCBF training approach proposed  in “Safe Control with Learned Certificates: A Survey of Neural Lyapunov, Barrier, and Contraction Methods for Robotics and Control” (Dawson et al, ref. [13] of the manuscript). We compared NCBFs with ReLU and tanh activation functions using the metrics (i)-(iii) above.
>
> The results are summarized in Table 1 of the pdf attachment to the general rebuttal. We found that, for the Darboux and obstacle avoidance case studies, the ReLU NCBF completed training faster than both sigmoid and tanh NCBFs. The volume of the safe region was comparable for all three activation functions, with the tanh outperforming the ReLU NCBF in Darboux and the ReLU NCBF providing the largest volume for obstacle avoidance. The most significant difference between the three activation functions was at the verification stage. Our proposed method for verifying ReLU NCBFs terminated within 15 and 274 seconds in the Darboux and obstacle avoidance, respectively, while SMT-based methods did not terminate within three hours for both test cases. In the spacecraft rendezvous example, the ReLU NCBF completed training before the tanh NCBF. Moreover, while our approach verified the correctness of the ReLU NCBF within 4 hours, the tanh NCBF exhibited a safety violation.
>
> Finally, to address the concern regarding high-dimensional problems, we evaluated our approach on an eight-dimensional system first defined in “Fossil: A Software Tool for the Formal Synthesis of Lyapunov Functions and Barrier Certificates Using Neural Networks” (Abate et al, ref. [1] in the manuscript). The results are summarized in Table 2 of the pdf attached to the general rebuttal. Our approach verified an NCBF with a single hidden layer containing 16 neurons in 35 seconds. The SOTA algorithms dReal and Z3 did not terminate within six hours.
>
> To address the concern on scalability to complex neural networks, we conducted additional experiments as shown in Tables 2 and 3 of the pdf attachment to the general rebuttal. We verified an NCBF for Darboux example with one hidden layer of 1024 neurons and an NCBF with two hidden layers of 512 neurons each. The verification terminated successfully in 108 and 620 seconds, respectively. The computational complexity of our approach will be determined by several factors including the dimension of the state, the number of layers, the number of neurons in each layer, and the geometry of the 0-level set of the NCBF. As shown in Table 3, the dimension of the system plays the most important role, which is a widely-shared issue in traditional SOS verification of CBFs as well as neural network verification algorithms. While the focus of this paper is on developing exact safety conditions, improving scalability is a direction of future work that we will pursue.

---

> > ### Comment · Reviewer_wvvT · 2023-08-13
> > **Response to rebuttal**
> >
> > I thank the authors for their response and conducting the additional experiments.
> > The majority of the concerns raised in my initial evaluation have been effectively attended to.
> > While I acknowledge the efforts undertaken by the authors to conduct an additional round of experiments, I still have reservations about how well the proposed algorithm would work for problems with high dimensions. The additional experiments provided are still confined to low-dimensional scenarios.
> > I recommend that the authors explore this aspect in their future research endeavors.

---

> > > ### Author Response · Authors · 2023-08-14
> > >
> > > We thank the reviewer for taking the time to give detailed feedback. We agree that scalability is a key challenge with neural network verification in general, and with verification of NCBFs in particular. Our current approach shows a significant improvement in runtime compared to the state of the art. Moreover, we believe that our approach can lead to future approaches to safety verification, e.g., by developing tractable sufficient but not necessary relaxations of the conditions derived in this paper. We plan to explore this aspect in future work as suggested by the reviewer.

---

### Official Review · Reviewer_5o49 · 2023-07-30

**Soundness:** 3 good
**Presentation:** 3 good
**Contribution:** 3 good
**Rating:** 6
**Confidence:** 3

**Summary:**

The paper proposes a technique to verify the safety of NCBF-based control policies. Usually, the verification depends on the CBF to be continuously differentiable. This is not the case for NNs, however using NCBFs is beneficial as it allows to encode more complex safety constraints.
Their approach identifies the piecewise linear segments of the NCBF. The number of segments is reduced by only focusing on those at the boundary of the safe region. The remaining linear segments are overapproximized and verified using nonlinear programs.
The authors compare their proposed technique with SOTA SMT based methods and demonstrate that they are able to verify NCBFs that previously resulted in a timeout.

**Strengths:**

The paper propose a new original technique to prove the safety of NCBFs.
The authors do a good job of motivating their new approach by demonstrating the shortcoming of techniques that expect $b$ to be continuously differentiable.
The paper is mostly clear in its explanation and formulas, and gives intuitive explanations for many of them.
In their experimental evaluation, they use one benchmark to compare their performance against two other SOTA techniques, and demonstrate that they are able to verify instances that would otherwise lead to a timeout. This indicates the significance of their proposed technique, should the results carry over to other benchmarks.

**Weaknesses:**

The equations in the paper get increasingly complicated to follow, even with the provided explanations. Especially Lemma 4 is hard to follow.

In the experimental evaluation, the paper would strongly benefit from exploring more benchmarks. The comparison to SOTA techniques dReal and Z3 is limited to one benchmark, with two more benchmarks that do not include a comparison to those tools. Also, a comparison to an approach based on neural-network-verification (e.g. "A Hybrid Partitioning Strategy for Backward Reachability of Neural Feedback Loops" by Nicholas Rober, Michael Everett, Songan Zhang, and Jonathan P. How) would help to demonstrate that the proposed technique can solve previously hard problems.

**Questions:**

Main question: Why can't one use regular verification of safety properties of neural networks? If $D$ is convex (or can be split into a reasonable number of convex subsets), then one could verify that no input in $D$ is mapped to an output outside of $D$. When $f$, $g$ and $\\mu$ in Equation 1 are known, that equation could most probably be encoded as a NNs, so all common tools (compare e.g. VNN-COMP 2022 or 2023) should be able to verify this (or time out). You state the main benefit of your approach is that it does not depend on the specific choice of $\\mu$ (line 288). However, I do not know if this is often a requirement. If $\\mu$ changes, the verification using a technique that depends on $\\mu$ could be repeated. Did you do an experimental comparison of your approach with a NN-Verification-Tool-based approach to see how costly this would be?

Other questions:

1) In the text above Proposition 1, is $\\overline{X}(S_1) \\cap \\ldots \\cap \\overline{X}(S_r) \\cap S'$ well-defined? $\\overline{X}(S)$ is a set of inputs $x$, but $S'$ is a set of neurons. Should this be $\\overline{X}(S')$?

2) In Equation 11, is the term to the right of $ \\textbackslash $ obsolete? If $\\{S_1, \ldots, S_r\\}$ is complete, then no input that activates all of $\\{S_1, \ldots, S_r\\}$ also activates any other $S'$ (based on the text above Proposition 1). So what is removed by the term right of $ \\textbackslash $ ?

3) What's the significance of the three different lines in Figure 2? They represent different "set boundaries", bu I do not know what that visualization achieves compared to one with just set boundary 0

Minor: What is the missing reference in line 243?

Minor note: Line 268: "with in" -> "within"

**Limitations:**

The authors do not provide a list of limitations or potential negative societal impact. However, the potential negative societal impact is probably small, as this technique is developed to increase the provable safety of NCBFs. So the potential negative societal impact is identical to that of any AI-based technology.
The paper would benefit from a description of the limitations of their proposed approach.

---

> ### Author Rebuttal · Authors · 2023-08-09
>
> Regarding the benchmarks considered in this work, we have evaluated our approach on additional test systems and compared with dReal and Z3. Our results are summarized in Tables 2 and 3 of the pdf attachment to the general rebuttal. Our approach verified a three-dimensional system (obstacle avoidance) with a 128-neuron NCBF in roughly 3 minutes, a six-dimensional system with an NCBF consisting of 102 neurons with three hidden layers within four hours, and an eight-dimensional system with an 18-neuron NCBF within 35 seconds. In contrast, the SOTA algorithms dReal and Z3 did not terminate within six hours for the obstacle avoidance and eight hours for the eight-dimensional system. All experiments were performed on the desktop PC environment described in our manuscript.
>
> We thank the reviewer for suggesting comparisons with the VNN-COMP benchmarks. VNN-COMP is primarily concerned with verifying input/output relationships of neural networks, i.e., proving that, given a neural network $b$ and a set of inputs $\mathcal{X}$, we have $b(\mathcal{X}) \subseteq \mathcal{Y}$ for some set $\mathcal{Y}$. In principle, it would be possible to train a neural network $\phi(x)$ to approximate $f(x)+g(x)\mu(x)$ as suggested by the reviewer, and then use the methodologies of VNN-COMP to check whether there exists $x$ with $b(x) \geq 0$ and $b(x+\phi(x)dt) < 0$, where $dt$ is a discrete-step size that approximates the evolution of the ODE (1). However, in order to achieve an exact verification algorithm, the errors introduced by the NN approximation $\phi(x)$ and the discrete-time approximation of (1) would need to be characterized and incorporated into the verification. Furthermore, as mentioned by the reviewer, our approach does not depend on the control policy $\mu(x)$. We believe that this is an advantage because we can ensure safety under any nominal control policy $\mu(x)$ by incorporating the policy into the optimization-based control (10). This provides the system designer with an additional degree of flexibility, which has been highlighted in recent works on CBF-based safety filters (e.g., A.D. Ames et al, “Control Barrier Functions: Theory and Applications”) and safe shield policies in reinforcement learning (e.g., I. ElSayed-Aly et al, “Safe Multi-Agent Reinforcement Learning via Shielding”).
>
> Similarly, we would like to highlight the following distinction with the related work “A Hybrid Partitioning Strategy for Backward Reachability of Neural Feedback Loops” by Rober et al. The main goal of this related work is to verify safety of a given neural network controller using backwards reachability analysis. In contrast, the goal of our work is to construct an NCBF b and prove that any control policy $\mu$ satisfying Eqs. (7)--(9) is safe. Our approach could be considered complementary to Rober et al in the following two ways. First, a given neural network controller $\mu$ could be modified to provide verifiable safety guarantees by following the quadratic program-based policy defined by Eq. (10) with $\mu_{nom} = \mu$. Second, one could attempt to prove that a NN feedback control policy satisfies Eqs. (7)--(9) for a given $b$ and all x, which would prove that the policy is safe. This latter approach to NN safety verification would be an alternative to backward reachability analysis, and is a direction of future research.
>
> Finally, we thank the reviewer for identifying several typos in the paper. Proposition 1 should be
> $\cdots \cap \overline{\mathcal{X}}(\mathbf{S}_{r}) \cap \overline{\mathcal{X}}(\mathbf{S}^{\prime})$
> as suggested by the reviewer. The term to the right of \ can indeed be omitted in Eq. (11). We have revised the visualization of Fig. 2 to remove the additional lines and thus improve readability. The missing reference in line 243 refers to Eq. (27) of the supplement and will be fixed. We will revise the paper carefully including Lemma 4 to simplify and clarify the notations.

---

> > ### Comment · Reviewer_5o49 · 2023-08-12
> >
> > Why would the construction of $\phi$ require a training? I'm not an expert in this field, so maybe I misunderstand what $\mu$ typically looks like. Could it easily be encoded by a neural network? E.g. $\mu(x) = \max(0, Mx)$ for some matrix $M$ could be encoded as one linear layer followed by a ReLU, without the need for any training. Then, $g(x)\phi(x)$ is a NN as well, and $f(x) + g(x)\phi(x)$ would be as well, using a residual connection. This would ensure that no errors are introduced.
> > I also do not understand the argument concerning the discrete-time approximation. Assuming a NN encoding $\phi$ can be constructed, shouldn't $x(t)$ be simply the output of the NN? I do not see how this introduces an additional error.
> >
> > Thank you for the additional experiments!

---

> > > ### Author Response · Authors · 2023-08-13
> > >
> > > We thank the reviewer for the comment. We would like to respond and clarify the model of Eq. (1) of the paper. First, there may be a straightforward neural network encoding of the right-hand side of (1) under certain cases, e.g., when $f$ is represented as a NN (or linear as a special case), $g$ is constant, and $\mu$ is linear or represented as a NN. However, for general nonlinear $f$, $g$, and $\mu$, e.g., when $f$ and $g$ contain polynomial or trigonometric functions of $x$ or the control $\mu$ is nonlinear, representing the right hand side of (1) as a NN will be nontrivial and may require training a NN $\phi$ as described in our rebuttal.
> > >
> > > Second, the left hand side of (1) is the time derivative $\dot{x}(t) = \frac{dx}{dt}$, not the state $x(t)$. Hence, applying the methods of VNN-COMP would involve checking whether there exists $x$ with $b(x) \geq 0$ and $b(\bar{x}) < 0$, where $\bar{x}$ is the forward integration of (1) over a time interval of length $dt > 0$ from initial state $x$. This forward integration operator would then need to be included in the formulation, for example, through the approximation $\bar{x} \approx x + dt\cdot\phi(x)$ described in the rebuttal.

---

### Author Rebuttal · Authors · 2023-08-09

We would like to thank the reviewers for providing detailed comments that have helped to improve the quality of our manuscript. We have provided rebuttals to the comments of each reviewer. We have also attached a pdf file containing figures and tables from additional simulations that were requested by the reviewers. In this general rebuttal, we briefly summarize these additional figures and tables. Figure 1 is in response to a comment from Reviewer yuYJ. The reviewer asked about the impact of non-differentiable points on safety filtering. We considered the example from Page 4 of the manuscript, and compared the unsafe NCBF from the manuscript (which we denote $b_{c}$) with a trained NCBF $b_{\theta}$ that we verified using our approach. We compared optimization-based controllers based on Eq. (10) of the manuscript using the NCBFs $b_{c}$ and $b_{\theta}$, where the nominal controller was a linear quadratic regulator that drives the system to the origin. The NCBF $b_{c}$ resulted in a safety violation due to the non-differentiable point identified in the manuscript, while the NCBF $b_{\theta}$ ensured that the system state remained within the safe region. Additional descriptions can be found in the rebuttal to Reviewer yuYJ.

Table 1 is in response to comments from Reviewers wvvT and hbj4. Both reviewers asked why ReLU activation function would be used for NCBFs instead of differentiable activation functions. To address this question, we considered three case studies, namely, Darboux, obstacle avoidance, and spacecraft rendezvous. Descriptions of each of these systems can be found in the manuscript. For each case study, we trained and verified three NCBFs with the same architecture (2 hidden layers of 32 neurons each) but different activation functions, namely, ReLU, sigmoid, and tanh. We found that the NCBF with ReLU activation function had the shortest training time for all test cases and resulted in the largest safe region volume for the obstacle avoidance test case. Moreover, our proposed verification algorithm for the ReLU NCBF terminated within 15, 274, and 13907 seconds for Darboux, obstacle avoidance, and spacecraft rendezvous, respectively, while the SOTA verification algorithms (dReal and Z3) did not terminate within three hours for any of the test cases.. Additional descriptions of these experiments can be found in the rebuttal to Reviewer wvvT.

Tables 2 and 3 are in response to comments from Reviewers 5o49, wvvT, U7BK, yuYJ, and hbj4 regarding the scalability of the proposed approach to higher-dimensional systems as well as neural networks with larger numbers of neurons. We trained NCBFs for an additional eight-dimensional system that first appeared in “Fossil: A Software Tool for the Formal Synthesis of Lyapunov Functions and Barrier Certificates Using Neural Networks” (Abate et al, ref. [1] in the manuscript), as well as more complex neural networks for the three case studies already considered in the paper. Our approach verified a 1024-neuron NCBF for the Darboux system in 620 seconds, a 128-neuron NCBF with two hidden layers for obstacle avoidance in 11749 seconds, a 96-neuron NCBF for spacecraft rendezvous in 13907 seconds, and a 16-neuron NCBF for the eight-dimensional test case in 35 seconds. Neither of the SOTA verification algorithms (dReal and Z3) terminated within three, six, and six hours for the Darboux, obstacle avoidance, and eight-dimensional systems, respectively. Additional details can be found in the rebuttals to Reviewers 5o49 and wvvT.

---

### Decision · Program_Chairs · 2023-09-21

**Decision:**

Accept (poster)

**Comment:**

During the initial phase of reviewing, some issues were raised regarding the clarity and experimental setup of the submission. The rebuttal provided the necessary clarifications. All five expert referees recommend acceptance. The authors should take into account the detailed reviews, discussions and their own rebuttal while preparing the camera-ready version of the paper.